# Promoting electrocatalytic $CO_2$ reduction to formate via sulfur-boosting water activation on indium surfaces

Wenchao Ma[1], Shunji Xie[1], Xia-Guang Zhang[1], Fanfei Sun[2], Jincan Kang[1], Zheng Jiang [2], Qinghong Zhang[1], De-Yin Wu [1] & Ye Wang [1]

Electrocatalytic reduction of $CO_2$ to fuels and chemicals is one of the most attractive routes for $CO_2$ utilization. Current catalysts suffer from low faradaic efficiency of a $CO_2$-reduction product at high current density (or reaction rate). Here, we report that a sulfur-doped indium catalyst exhibits high faradaic efficiency of formate (>85%) in a broad range of current density (25–100 mA $cm^{-2}$) for electrocatalytic $CO_2$ reduction in aqueous media. The formation rate of formate reaches 1449 $\mu$mol $h^{-1}$ $cm^{-2}$ with 93% faradaic efficiency, the highest value reported to date. Our studies suggest that sulfur accelerates $CO_2$ reduction by a unique mechanism. Sulfur enhances the activation of water, forming hydrogen species that can readily react with $CO_2$ to produce formate. The promoting effect of chalcogen modifiers can be extended to other metal catalysts. This work offers a simple and useful strategy for designing both active and selective electrocatalysts for $CO_2$ reduction.

[1] State Key Laboratory of Physical Chemistry of Solid Surfaces, Collaborative Innovation Center of Chemistry for Energy Materials, National Engineering Laboratory for Green Chemical Productions of Alcohols, Ethers and Esters, College of Chemistry and Chemical Engineering, Xiamen University, 361005 Xiamen, China. [2] Shanghai Synchrotron Radiation Facility, Shanghai Institute of Applied Physics, Chinese Academy of Sciences, 201204 Shanghai, China. These authors contributed equally: Wenchao Ma, Shunji Xie, Xia-Guang Zhang. Correspondence and requests for materials should be addressed to Q.Z. (email: zhangqh@xmu.edu.cn) or to D.-Y.W. (email: dywu@xmu.edu.cn) or to Y.W. (email: wangye@xmu.edu.cn)

Catalytic transformations of $CO_2$ to fuels and chemical feedstocks contribute to establishing carbon-neutral cycle to alleviate the rapid consumption of fossil resources and the growing emission of $CO_2$[1–4]. The electrocatalytic reduction of $CO_2$ has become one of the most attractive routes for $CO_2$ transformations owing to recent progress in generating electricity from renewable energy sources such as solar and wind[5,6]. Formate, which is widely used as a feedstock in pharmaceutical, tanning and textile industry, and can also be a hydrogen carrier for fuel cell[7,8], is a very attractive $CO_2$-reduction product. The electrocatalytic reduction of $CO_2$ to formate is also economically viable considering the energy input and the market value of product[9]. Many studies have been devoted to the electrocatalytic reduction of $CO_2$ to formate[10–13], but no catalyst can work with high activity, selectivity and stability. The development of highly efficient electrocatalysts for $CO_2$ reduction to formate to meet the commercial purpose remains challenging.

Metal catalysts have typically been employed in electrocatalytic $CO_2$ reduction reaction ($CO_2RR$) because of high activity and stability[10,13–16]. Accompanying with $CO_2RR$ to formate (Eq. 1), the hydrogen evolution reaction (HER) (Eq. 2) also occurs as a competitive reaction. The inhibition of HER is essential in obtaining high $CO_2RR$ selectivity and formate selectivity. The catalysts with high $CO_2RR$ selectivity are the metals typically located at the left-hand branch of Trassati's volcano curve[17], such as Ag[14], Zn[15], Pb[16], Sn[16], and In[10], having weak metal-hydrogen bond and low HER activity.

$$CO_2 + H_2O + 2e^- \rightarrow HCOO^- + OH^- \tag{1}$$

$$2H_2O + 2e^- \rightarrow H_2 + 2OH^- \tag{2}$$

Recent studies showed that oxide-derived and sulfide-derived metals had better electrocatalytic $CO_2RR$ performances as compared to the corresponding pure metal catalysts probably because of the unique surface structures and local environments such as roughness, defects and oxygen (or sulfur) modifiers[11,12,18–22]. Although high $CO_2RR$ selectivity has recently been achieved over some catalysts, the selectivity of formate is sensitive to the applied potential or the current density (Supplementary Table 1)[11,12,18–22]. Faradaic efficiency (FE) of formate drops at a high current density (>60 mA cm$^{-2}$) because of the significant enhancement in HER. This results in limited formation rate of formate (<1000 µmol h$^{-1}$ cm$^{-2}$) (Supplementary Table 1). Therefore, it would be a significant step forward to develop an effective strategy to accelerate the activity while keeping the high formate selectivity.

Indium, a non-noble metal, has emerged as a $CO_2RR$ catalyst for selective formation of formate with high FE (≥75%). However, the activity of indium catalysts is usually low (current density <6 mA cm$^{-2}$),[10,23,24] although the design of special electrochemical cell can enhance the activity[25]. Here, we report a unique sulfur-doped oxide-derived indium catalyst, which not only shows high $CO_2RR$ activity and FE of formate but also can keep high FE of formate in a large range of current density. The formation rate of formate of our catalyst breaks the current upper limit of 1000 µmol h$^{-1}$ cm$^{-2}$. We discovered that the presence of sulfur accelerates the activation of water. The unique hydrogen species thus formed unexpectedly enhances electrocatalytic $CO_2RR$ to formate instead of HER. This offers an effective strategy to develop superior $CO_2RR$ electrocatalysts with not only high selectivity but also high activity.

## Results
**$CO_2RR$ performances of sulfur-doped indium catalysts**. Sulfur-doped indium (denoted as S−In) catalysts were fabricated by electroreduction of sulfur-containing $In_2O_3$ precursors, which grew on carbon fibers by a solvothermal method. The obtained catalysts with sulfur contents of 0, 2.5, 4.9, 9.4, and 14 mol%, which were determined by Auger electron spectroscopy (AES) (Supplementary Fig. 1), were denoted as S0–In, S1−In, S2−In, S3−In, and S4−In, respectively. The electrocatalytic study showed that our $In_2O_3$-derived metal catalyst on carbon fibers exhibited higher activity for $CO_2RR$ to formate than the commercial In foil at a potential of −0.98 V versus reversible hydrogen electrode (RHE) (Fig. 1a). The formation rate of formate increased significantly with an increase in sulfur content up to 4.9 mol% (S2−In), whereas the formation rates of $H_2$ and CO only changed slightly at the same time. The FE of formate also increased with sulfur content. A further increase in sulfur content to >4.9 mol% rather decreased the formation rate of formate. Thus, the best performance was achieved over the S2−In catalyst. The formation rate and FE of formate over the S2−In catalyst reached 1002 µmol h$^{-1}$ cm$^{-2}$ and 93% at −0.98 V versus RHE, respectively, which were about 17 and 1.6 times those over In foil.

We conducted $^{13}CO_2$ labeling experiments for the S2−In catalyst. The products obtained at a potential of −0.98 V versus RHE were analyzed by $^1H$ and $^{13}C$ nuclear magnetic resonance (NMR) spectroscopy. A $^1H$ NMR doublet was observed at 8.5 ppm, which was attributable to the proton coupled to $^{13}C$ in $H^{13}COO^-$ (Supplementary Fig. 2a). A signal at 168.5 ppm was observed in the $^{13}C$ NMR spectrum, which could be ascribed to $H^{13}COO^-$ (Supplementary Fig. 2b)[11]. These observations confirm that formate is formed from $CO_2$ reduction.

We performed further studies for the most efficient S2−In catalyst at different cathodic potentials. The $CO_2RR$ started to occur at a potential of −0.33 V versus RHE (overpotenial, 0.14 V) with FE of formate of 3% (Supplementary Fig. 3a), comparable to that over a partially oxidized Co catalyst[11]. Eighty percent FE of formate was achieved at −0.63 V versus RHE (overpotential, 0.44 V), better than those over most of the non-noble catalysts under such a lower overpotential (Supplementary Table 1). The change in the applied potential from −0.33 to −1.23 V versus RHE resulted in a variation in current density in a broad range from 0.15 to 100 mA cm$^{-2}$, and the current density kept stable during the electrocatalysis at each given potential (Fig. 1b). The current density ascribed to $CO_2RR$, which was calculated by considering the FE of $CO_2RR$, increased significantly from 0.03 to 86 mA cm$^{-2}$ by changing potential from −0.33 to −1.23 V versus RHE and then became almost saturated (Supplementary Fig. 4). It is noteworthy that the current density of $CO_2RR$ of 86 mA cm$^{-2}$ approaches the maximum value (90 mA cm$^{-2}$) evaluated by assuming the mass-transport limitation under our reaction conditions (Supplementary Methods). The performance of the S2−In catalyst was further compared with that of In foil, a reference catalyst, at different potentials. The S2−In catalyst exhibited significantly higher current density, FE and formation rate of formate than In foil at each potential (Supplementary Fig. 3a–c). For a better comparison, we have normalized the formation rate of formate based on the electrochemical surface area (ECSA) (Supplementary Table 2), which was determined by double-layer capacitance ($C_{dl}$) method (Supplementary Fig. 5). The S2−In catalyst exhibited higher normalized formation rate of formate than In foil (Fig. 1c). The superiority of the S2−In catalyst for the formation of formate became less significant at potentials more negative than −1.03 V versus RHE probably because of the mass-transport limitation at a high current density.

It is quite unique that the high FE of formate (>85%) can be maintained in a large range of current density (25–100 mA cm$^{-2}$) over the S2−In catalyst (Supplementary Figs. 3a and 1d). For comparison, the behaviors of some typical catalysts, which have been reported as top $CO_2RR$-to-formate catalysts, are also

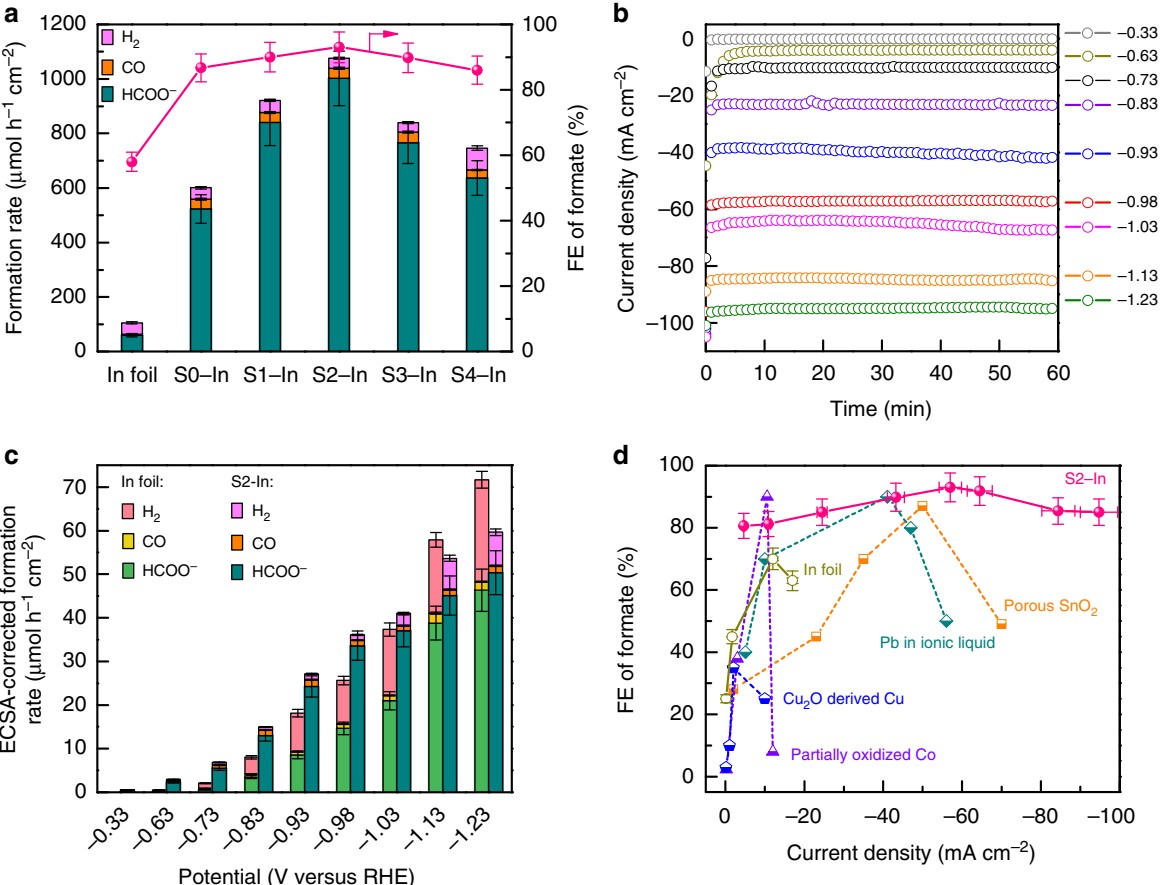

**Fig. 1** CO$_2$RR performances of sulfur-doped indium catalysts. **a** Formation rates of H$_2$, CO and HCOO$^-$ and FE of formate for In foil and S−In catalysts at −0.98 V (versus RHE) for 1 h. **b** Current density for S2−In catalyst over 1 h of reaction at each given potential (versus RHE). **c** ECSA-corrected current density and FE of formate for In foil and S2−In catalyst at each given potential for 1 h. **d** Plot of FE of formate versus current density for S2−In catalyst and some typical catalysts reported to date (see Supplementary Table 1 for details). Reaction conditions: CO$_2$-saturated 0.5 M KHCO$_3$ solution in H-type electrochemical cell with platinum plate as the counter electrode and saturated calomel electrode (SCE) as the reference electrode. The experiments in each case were performed at least for three times. The error bar represents the relative deviation

displayed in Fig. 1d. The FE of formate drops significantly to <60% at a current density of >60 mA cm$^{-2}$ over all the electrocatalysts reported to date even in ionic liquid or organic electrolyte (Supplementary Table 1). Furthermore, the S2−In catalyst showed excellent stability in 10 h operation (Supplementary Fig. 6). All these facts demonstrate that the present S2−In catalyst, which shows high selectivity at high current density and thus high reaction rate, is very promising for CO$_2$RR to formate.

**Characterizations of sulfur-doped indium catalysts**. The X-ray diffraction (XRD) patterns confirmed that In$_2$O$_3$ was the only crystalline phase in precursors and In$_2$O$_3$ was reduced to metallic In after electroreduction (Supplementary Fig. 7). Only diffraction peaks ascribed to metallic In with tetragonal phase could be observed for the S−In catalysts with different sulfur contents. The scanning electron microscopy (SEM) measurements showed that In particles were uniformly distributed on carbon fibers in each sample (Fig. 2a and Supplementary Fig. 8a–e). The average diameters of In particles were evaluated to be similar (110–131 nm) in the S−In samples with different sulfur contents (Supplementary Fig. 9a–e). After electrocatalytic reaction, the mean size of In particles in the S2−In catalyst maintained almost unchanged (Supplementary Figs. 8f and 9f). The high-resolution

transmission electron microscopy (HRTEM) measurements for the S−In samples displayed lattice fringes with an interplanar spacing of 0.272 nm (Fig. 2a and Supplementary Fig. 10), which could be ascribed to the In (101) facet. The catalyst loading on carbon fibers was 0.5 ± 0.1 mg cm$^{-2}$ for each catalyst. These suggest that there are no significant differences in non-chemical parameters for the S−In series of catalysts, such as catalyst loading, size or dispersion of In particles and catalyst porosity. Thus, these parameters do not account for the enhanced current density and the formation rate of formate after the modification of In catalysts by sulfur.

The energy-dispersive X-ray spectroscopy (EDS) analysis for the S2−In catalyst indicated that In, S and O elements existed in the catalyst, and these elements were distributed uniformly over the catalyst particle (Fig. 2b and Supplementary Fig. 11). The sputtering of the S2−In sample with Ar ions resulted in a significant decrease in the signal of S but the signal of In rather increased slightly in the AES spectra (Supplementary Fig. 12). This observation suggests that sulfur species are mainly located on the surface of In particles. The X-ray absorption near-edge structure spectroscopy (XANES) and X-ray photoelectron spectroscopy (XPS) were used to investigate the chemical states of indium and sulfur. The XANES measurements for the S2−In catalyst before and after electrocatalytic reaction displayed the same pattern in In K-edge (Fig. 2c), indicating that the chemical state of indium did

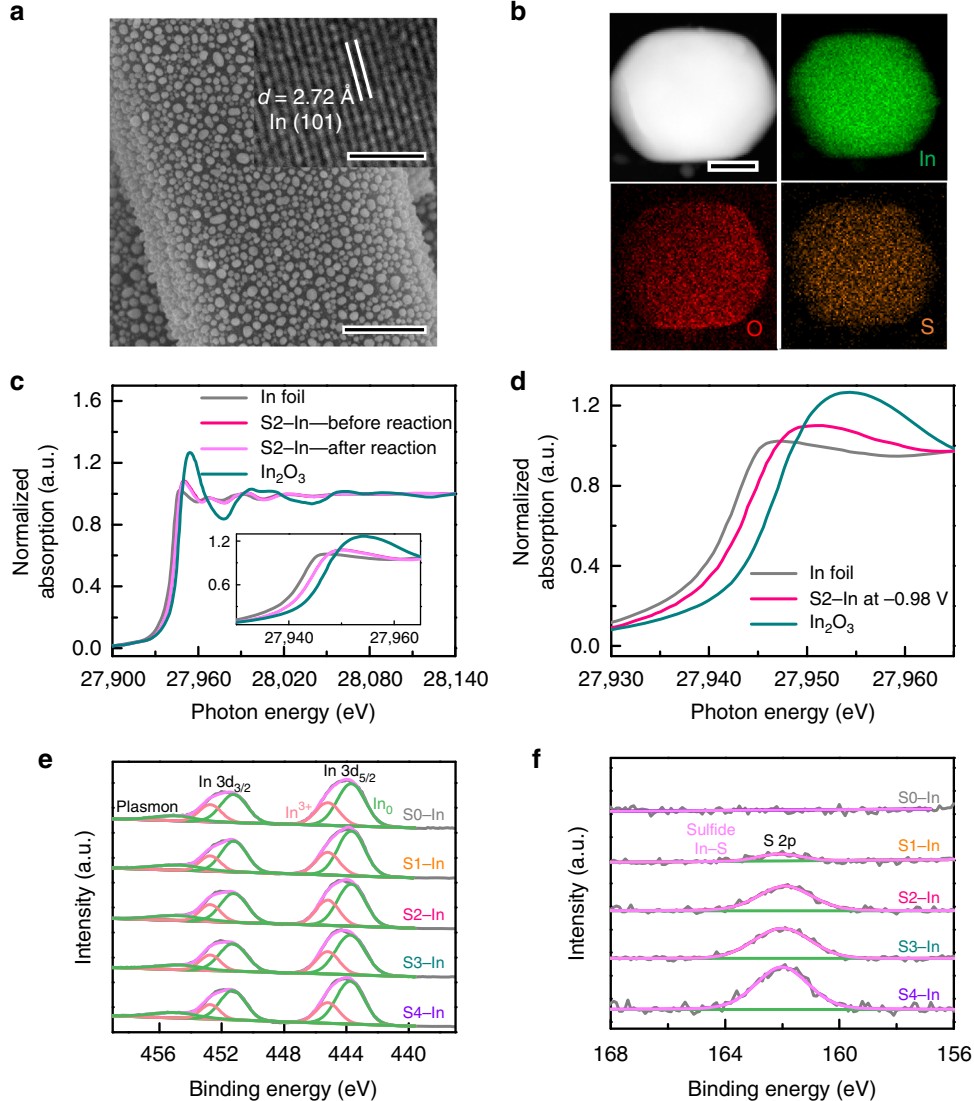

**Fig. 2** Characterizations of morphologies and chemical states for S—In catalysts. **a** SEM image and HRTEM image (insert) of S2—In catalyst. **b** STEM image of S2—In catalyst and the corresponding EDS elemental mapping. **c** In K-edge XANES spectra for S2—In catalyst before and after reaction. **d** In situ In K-edge XANES spectra for S2—In catalyst at −0.98 V versus RHE. **e** and **f** In 3d and S 2p XPS spectra of S—In catalysts

not change before and after reaction. The comparison of the pattern with those for reference samples, i.e., In foil and $In_2O_3$, suggests that the oxidation state of indium in the S2—In catalyst before and after reaction lies between 0 and +3. We further performed in situ XANES measurements for the S2—In catalyst under electrocatalytic $CO_2RR$ at −0.98 V (versus RHE). The result indicates that indium is also in the oxidation state between $In^0$ and $In^{3+}$ under reaction conditions (Fig. 2d). The In $3d_{5/2}$ and In $3d_{3/2}$ spectra obtained from XPS measurements could be deconvoluted into $In^0$ and $In^{3+}$ components for the S—In samples with different sulfur contents (Fig. 2e)[23]. This further suggests that $In^0$ and $In^{3+}$ species co-exist on the surfaces of S—In catalysts. The S 2p spectra for the S—In catalysts displayed a peak at 161.8 eV, which could be assigned to $S^{2-}$ in sulfides (Fig. 2f)[26]. The O 1 s spectra displayed a peak at 530.6 eV, which could be assigned to $O^{2-}$ in $In_2O_3$ (Supplementary Fig. 13)[23,27]. Thus, $In_2O_3$ species co-exist with metallic In on the catalyst surface in addition to sulfide species. The XPS results for the S2—In catalyst after electrocatalytic $CO_2RR$ reaction revealed that the surface states of indium, sulfur and oxygen did not undergo significant changes during the electrocatalysis (Supplementary Fig. 14).

Our electrochemical characterizations clarified that the ECSA for the S—In series of catalysts did not change significantly with sulfur content (Supplementary Table 2). Thus, as mentioned before, the enhancing effect of sulfur is not active-surface-area related. The linear sweep voltammetry and electrochemical impedance spectra measurements in $CO_2$-saturated 0.5 M KHCO$_3$ aqueous solution showed that the presence of sulfur on indium increased the cathodic current density and accelerated the charge-transfer kinetics in the electrocatalysis (Supplementary Fig. 15).

**Functioning mechanism of sulfur and effects of indium state.** Our present work has demonstrated that the sulfur-modified In catalyst is very promising for electrocatalytic reduction of $CO_2$ to formate. To understand the role of sulfur more deeply, it is necessary to disentangle different factors that may contribute to $CO_2RR$ in the present system. Our results show that the S0—In catalyst without sulfur fabricated by electroreduction of $In_2O_3$ precursor growing on carbon fibers exhibits higher FE of formate than In foil (Fig. 1a and Supplementary Fig. 16a). The activity of

the S0−In catalyst, expressed by the ECSA-corrected formation rate of all products (including $HCOO^-$, CO, and $H_2$), is almost the same with that of In foil (Supplementary Fig. 16b). The S0−In catalyst exhibits nanoparticulate morphology with an average diameter of 128 nm, whereas In foil has smooth surfaces (Supplementary Fig. 8a and 8g). Moreover, our XPS measurements reveal that a small fraction of $In^{3+}$ (i.e., $In_2O_3$) species co-exists with metallic In on the S0−In surface. The nanostructured morphology and the presence of oxidized species on metal catalysts were reported to be beneficial to $CO_2RR$[23,28–30]. In particular, $In(OH)_3$ was proposed to play a crucial role in the formation of formate or CO[23,30]. Our XPS results indicated the co-existence of $In_2O_3$ but not $In(OH)_3$ with $In^0$ in our case. To understand the role of surface oxidized species, we further pre-treated In foil in air at 250 °C for 3 h to generate a coverage of $In_2O_3$ on In surfaces. The electrocatalytic $CO_2RR$ result showed that the FE of formate increased on the surface-oxidized In foil, although the formation rate of all products based on ECSA did not change significantly (Supplementary Fig. 16). We performed $CO_2$ adsorption under gas-phase conditions to compare the $CO_2$ adsorption capacity among different catalysts. Our measurements revealed that the ECSA-corrected $CO_2$ adsorption amount increased in the sequence of In foil < surface-oxidized In foil < S0−In (Supplementary Fig. 17), and this agrees with the sequence of FE of formate. Therefore, we propose that the co-existence of oxide species, as well as the nanostructure morphology may account for the high FE of formate during the $CO_2RR$ over the S0−In catalyst probably by enhancing the adsorption of $CO_2$ onto catalyst surfaces.

To demonstrate the intrinsic role of sulfur, we have modified the S0−In catalyst with sulfur by a simple impregnation method. The obtained S-impregnated S0−In catalysts with sulfur contents ranging from 0 to 7.1 mol% have been used for $CO_2RR$. The formation rate of formate increased with an increase in sulfur content up to 2.6 mol% and then decreased (Supplementary Fig. 18a). We performed similar studies using In foil to further exclude the influences of nanostructures and surface oxide species. The electrocatalytic $CO_2RR$ using S-impregnated In foil catalysts with sulfur contents of 0–7.0 mol% showed similar dependences of catalytic behaviors on sulfur content (Supplementary Fig. 18b). The presence of sulfur on In foil with a proper content (≤2.2 mol%) significant enhanced the formation rate of formate, although the value of formation rate was much lower as compared with that on the S-impregnated S0−In series of catalysts. The change in FE of formate with sulfur content was less significant for both series of catalysts (Supplementary Fig. 18). These results are consistent with those observed for the S−In series of catalysts (Fig. 1a) and confirm that the sulfur species on In surfaces contributes to promoting the activity of $CO_2RR$ to formate.

Moreover, when we added a small amount of $Zn^{2+}$ to block the surface $S^{2-}$ sites through the strong interaction between $Zn^{2+}$ and $S^{2-}$ sites[31], the formation rate of formate over the S2−In catalyst decreased from 1002 to 687 µmol $h^{-1}$ $cm^{-2}$ (Supplementary Table 3). This observation provides further evidence that it is the $S^{2-}$ species but not other factors that plays a key role in accelerating the $CO_2RR$ to formate over the S−In catalysts.

As mentioned above, the enhancement in the adsorption and activation of $CO_2$ is vital for obtaining high $CO_2RR$ performance. However, our results reveal that the presence of sulfur does not significantly promote $CO_2$ adsorption (Supplementary Fig. 17). We propose that the sulfur species may enhance the $CO_2RR$ to formate by accelerating the activation of water. As shown in Eq. 1, the reduction of $CO_2$ to formate also consumes $H_2O$, but so far the activation of $H_2O$ has been overlooked in the $CO_2RR$. In particular, the activation of $H_2O$ in alkaline media is a slow step,

which even determines the kinetics of $H_2$ evolution reaction (HER)[32,33]. It is reported that the $H_2$ formation activity is one order of magnitude lower under alkaline conditions (pH = 13) than that in an acid electrolyte (pH = 1) during the HER over Au(111) surfaces[33], because of the difficulty in the reduction of water in alkaline electrolyte as compared to the discharging of hydronium in acid electrolyte. The alkaline electrolyte is widely employed in literature for $CO_2RR$ and also in our work. We consider that the activation of $H_2O$ would also be a slow step for $CO_2RR$ in alkaline medium.

To gain further insights into the role of the activation of $H_2O$ in $CO_2RR$, we have conducted studies on the kinetic isotope effect (KIE) of H/D over the S2−In catalyst. When $D_2O$ was used to replace $H_2O$ in 0.5 M $KHCO_3$ electrolyte, the formate formed was almost in the form of $DCOO^-$ (538 µmol $h^{-1}$ $cm^{-2}$) instead of $HCOO^-$ (10 µmol $h^{-1}$ $cm^{-2}$) (Supplementary Fig. 19). This indicates that the hydrogen in formate mainly originates from water rather than $HCO_3^-$. The KIE of H/D in $CO_2RR$ to formate was calculated to be 1.9. This KIE value is characteristic of primary kinetic isotopic effect[34]. We have also measured KIE of H/D in 0.5 M $K_2SO_4$ electrolyte and obtained the same result (Supplementary Fig. 19). This result provides evidence that the dissociation of water is involved in the rate determining step for $CO_2RR$ to formate over our S−In catalysts.

In our system, in $N_2$-saturated 1.0 M KOH solution without $CO_2$, the formation rate of $H_2$ was also found to increase with an increase in sulfur content in the series of S−In catalysts or in the S-impregnated S0−In and In foil catalysts (Supplementary Fig. 20). Several recent studies have also reported the role of adsorbed anions including $S^{\delta-}$ species on metal surfaces in accelerating the activation of $H_2O$ in alkaline media[31–33]. It is proposed that $S^{\delta-}$−hydrated cation $(K^+(H_2O)_n)$ networks can be formed in the double layer through non-covalent Coulomb interactions between the surface anionic sulfur species and the hydrated cation. This can promote the dissociation of $H_2O$ to form adsorbed hydrogen intermediate (H*), i.e., the Volmer step $(2H_2O + M + 2e^- \rightarrow 2M-H^* + 2OH^-)$, which is believed to be a slow step in HER[31–33]. However, to the best of our knowledge, there is no report to correlate the $CO_2RR$ activity with the enhancement in the activation of $H_2O$, because the current consensus is the enhancement in HER would decrease the $CO_2RR$ selectivity.

To obtain further evidence for the role of $H_2O$ activation in $CO_2RR$, we have investigated the effect of pH of electrolyte on electrocatalytic $CO_2RR$ over S0−In and S2−In catalysts. Three different electrolytes, i.e., $K_2HPO_4$, $KHCO_3$, and $K_2SO_4$, were employed to regulate the pH value, because it is known that the local pH at the cathode/electrolyte interface increases in the following sequence: $K_2HPO_4 < KHCO_3 < K_2SO_4$[35,36]. Our electrocatalytic results show that the formation rate and FE of formate increase in the sequence of $K_2HPO_4 < KHCO_3 < K_2SO_4$ over both catalysts (Supplementary Fig. 21a and 21b), indicating that a higher local pH environment favors the formation of formate. As compared to the S0−In, the S2−In catalyst exhibited higher formation rate and FE of formate using all the three electrolytes. Furthermore, the ratio of formation rates of formate for the S2−In and S0−In catalysts, i.e., $Rate_{S2−In}/Rate_{S0−In}$, increased from 1.4 to 1.9 and further to 2.1 upon changing the electrolyte from $K_2HPO_4$ to $KHCO_3$ and further to $K_2SO_4$ (Supplementary Fig. 21a). This suggests that the role of sulfur in enhancing the formation of formate is more significant at a higher pH value. This supports our speculation that the sulfur modification enhances formate formation by accelerating the activation of $H_2O$, which becomes more difficult at a higher pH[32,33].

To obtain further information on the promoting effect of sulfur on indium surfaces, we have performed density functional theory

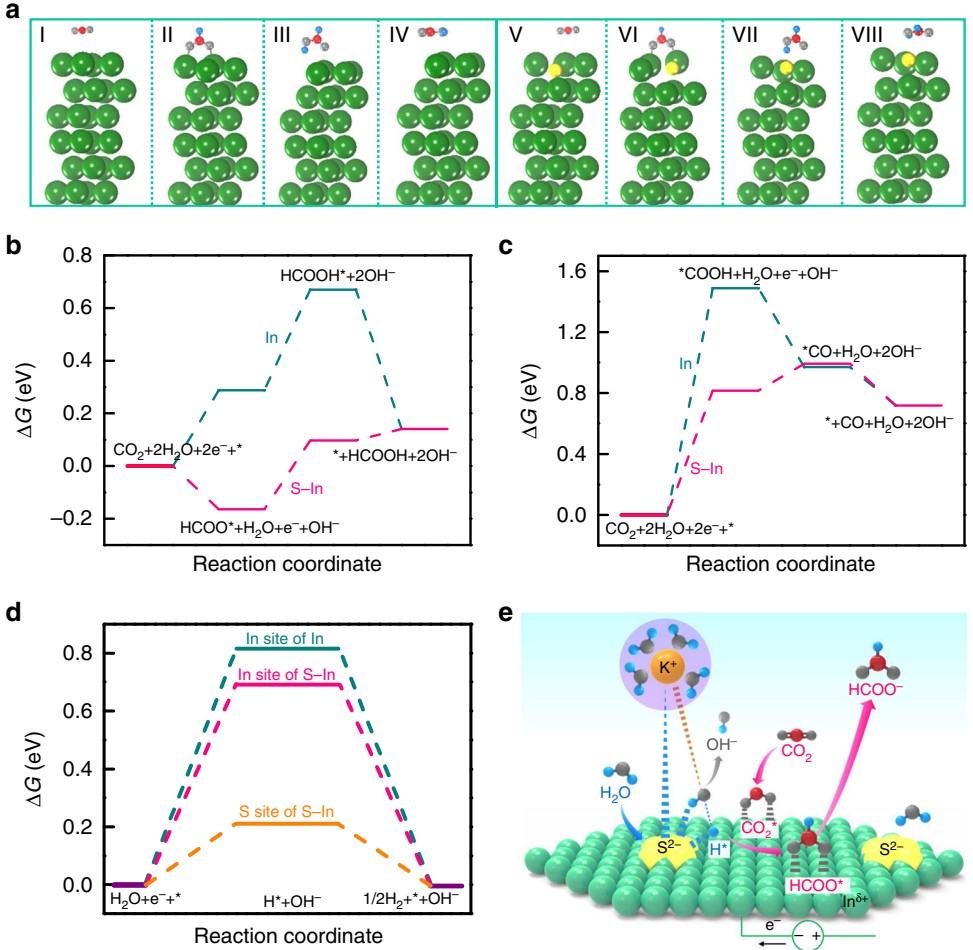

**Fig. 3** DFT calculation results and reaction scheme. **a** Optimized configurations of I $CO_2$, II HCOO*, III HCOOH*, IV HCOOH on (101) facet of pure indium (In) and V $CO_2$, VI HCOO*, VII HCOOH*, VIII HCOOH on (101) facet sulfur-doped indium (S–In). **b** Gibbs free energy diagrams for $CO_2$RR to HCOOH on In (101) and S–In (101) surfaces. **c** Gibbs free energy diagrams for $CO_2$RR to CO on In (101) and S–In (101) surfaces. **d** Gibbs free energies for the formation of H* on pure In (101), In and S sites of S–In (101) surfaces. **e** Schematic illustration for the role of $S^{2-}$ in promoting water dissociation and H* formation for the reduction of $CO_2$ to formate. Free energies of **b**, **c** and **d** are shown relative to gas $CO_2$ and $H_2$. The green, yellow, gray, red, and blue balls represent In, S, O, C, and H

(DFT) calculations for $CO_2$RR to HCOOH and CO on indium sole and sulfur-doped indium surfaces, and the results are summarized in Supplementary Table 4. The optimized adsorption configurations of reactants, intermediates and products on indium and sulfur-doped indium surfaces are displayed in Fig. 3a and Supplementary Fig. 22. The activation of $CO_2$ occurs on indium sites and the transfer of a proton/electron pair or adsorbed H intermediate to $CO_2$ leads to the formation of bound formate intermediate (HCOO*) on two indium sites via two oxygen atoms (Fig. 3a) or bound carboxylate intermediate (*COOH) on single indium site via carbon atom (Supplementary Fig. 22). HCOO* and *COOH are believed to be intermediates for the formations of HCOOH and CO, respectively[37–39]. For the HCOOH pathway, the Gibbs free energies ($\Delta G$) for the formations of HCOO* and HCOOH* are 0.29 and 0.67 eV, respectively on indium only surfaces (Fig. 3b). The presence of sulfur on indium significantly decreases the corresponding Gibbs free energies for HCOO* and HCOOH* to −0.16 and 0.10 eV, respectively (Fig. 3b). These results suggest that the doping of sulfur on indium surfaces makes the HCOOH pathway significantly energy-favorable. For the CO pathway, the Gibbs free energies for the formation of *COOH are 1.49 and 0.82 eV on pure and sulfur-doped indium surfaces, respectively (Fig. 3c). Thus,

the HCOOH pathway is more energy-favorable than the CO pathway, and thus can interpret why both pure and sulfur-doped indium surfaces possess higher selectivity of HCOOH than that of CO.

We have further calculated the Gibbs free energies for the HER in the absence of $CO_2$ on pure In and sulfur-doped In surfaces. The formation energy of H* species is 0.21 eV on sulfur sites of sulfur-doped In, significantly lower than that on In sites of sulfur-doped In (0.69 eV) and pure In (0.82 eV) (Fig. 3d). The lower formation energy of H* species means a higher activity of $H_2O$ dissociation on the electrocatalyst surface[40–43]. Therefore, our calculation results indicate that the sulfur modification can enhance the HER and the sulfur site on In surfaces is responsible for the dissociation of $H_2O$ to form the adsorbed H* intermediate. On the other hand, in the presence of $CO_2$, our DFT calculation reveals that the doping with sulfur has turned the formation of HCOO*, the precursor of formate, on the S–In surface to be exergonic (Fig. 3b), whereas the formation of H* from $H_2O$ alone, still remain endergonic. We believe that this is the major reason for why the formation of formate but not the formation of $H_2$ is preferentially enhanced in the case of $CO_2$RR after the modification of In by sulfur (Fig. 1a), although sulfur can boost the activation of $H_2O$.

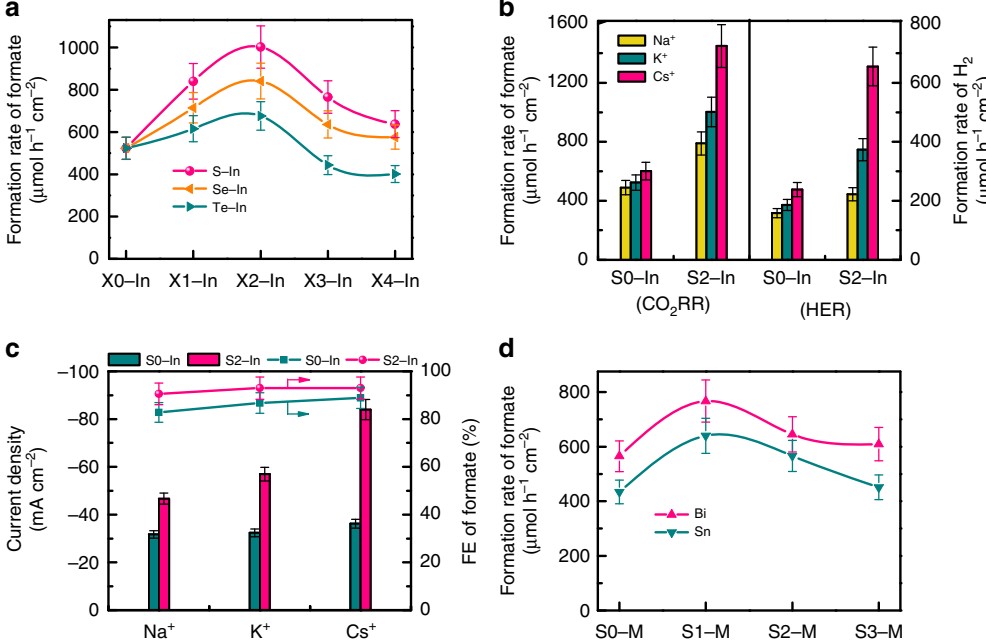

**Fig. 4** Several promoted metal-catalyzed $CO_2RR$ systems. **a** Formation rates of formate over S–In, Se–In and Te–In catalysts at −0.98 V (versus RHE) for 1 h. **b** Effect of alkali metal cations ($Na^+$, $K^+$, and $Cs^+$) in $MHCO_3$ (for $CO_2RR$) or MOH (for HER) electrolyte on $CO_2RR$ and HER performances for S0−In and S2−In catalysts at −0.98 V (versus RHE) for 1 h. **c** Effect of alkali metal cations ($Na^+$, $K^+$, and $Cs^+$) in $MHCO_3$ for $CO_2RR$ on average current density and FE of formate at −0.98 V (versus RHE) for 1 h. **d** Formation rates of formate over S−Bi and S−Sn catalysts at −0.98 V (versus RHE) for 1 h. The experiments in each case were performed at least for three times. The error bar represents the relative deviation

On the basis of the results and discussion described above, we propose that the surface $S^{2-}$ species serves as an anchor to keep the $K^+(H_2O)_n$ cation close to indium surfaces in the double layer via Coulomb interactions (Fig. 3e). The near-surface $H_2O$ molecules can be activated facilely, forming adsorbed $H^\star$ intermediate and releasing an $OH^-$ anion. The $H^\star$ intermediate can subsequently react with the adsorbed $CO_2$ to form a bound $HCOO^\star$ intermediate. After accepting an electron, $HCOO^\star$ is transformed to formate and desorbs from indium surfaces. These proposed elementary steps are displayed in Supplementary Note 1.

It is noteworthy that Pérez-Ramírez and co-workers recently reported a promotion effect of sulfur modification on the reduction of $CO_2$ to formate over Cu catalyst[21,44]. The doping of sulfur mainly changed the product selectivity and the FE of formate increased from 26% to 78% after sulfur modification over the Cu catalyst. The sulfur adatom on Cu surfaces is proposed to participate actively in $CO_2RR$ as a nucleophile either by transferring a hydride or by tethering $CO_2$, thus suppressing the formation of CO.[44] The different behaviors of sulfur doping on In and Cu catalysts reveal diversified functioning mechanisms of sulfur for $CO_2RR$.

**Generality of chalcogenide-modified metal-catalyzed $CO_2RR$.** Platinum is a powerful HER catalyst with strong ability for the formation of adsorbed atomic hydrogen species[32,41], and thus Pt might also work as a promoter for $CO_2RR$ according to our hypothesis. We found that the doping of small amount of Pt onto indium could promote the formation of formate to some extent (Supplementary Fig. 23a), but the FE of formate decreased because the formations of $H_2$ and CO were accelerated more significantly (Supplementary Fig. 23b). Pt not only enhances the formation of adsorbed $H^\star$ species but also accelerates the recombination of $H^\star$ to $H_2$, and thus is not a good promoter for $CO_2RR$ to formate. On the other hand, sulfur accelerates the

activation of $H_2O$ without significantly enhancing the recombination of $H^\star$ intermediates to $H_2$.

Indium catalysts doped with other chalcogen species have also been investigated for the $CO_2RR$. A series of selenium-doped and tellurium-doped indium catalysts, denoted as Se−In and Te−In, were fabricated by a similar method to that for the S−In catalysts. The electrocatalytic results showed that the doping of a proper amount of selenium or tellurium could also promote the formation of formate (Fig. 4a). The FE of formate also slightly increased by the modification of In with Se or Te (Supplementary Fig. 24a), suggesting that selenium or tellurium modifier played similar roles to sulfur. The formation rate of formate decreased in the sequence of S−In > Se−In > Te−In (Fig. 4a). The decrease in the electronegativity along the chalcogen group from sulfur to tellurium would decrease the interaction between chalcogenide and the hydrated cation[43], and thus would lower the ability to activate $H_2O$ to adsorbed $H^\star$ species.

The $K^+$ cation in the electrolyte could be replaced by other alkali metal cations, but the $CO_2RR$ performance was affected by the metal cation employed. Upon changing the cation from $Na^+$ to $K^+$ and further to $Cs^+$, the formation rate of formate increased significantly from 789 to 1002 and further to 1449 µmol $h^{-1}$ $cm^{-2}$ at a potential of −0.98 V versus RHE over the S2−In catalyst (Fig. 4b). The current density increased from 47 to 57 and further to 84 mA $cm^{-2}$ at the same time, while the FE of formate kept at 91–93% (Fig. 4c). On the other hand, the change in the reaction rate was very limited over the S0−In catalyst without sulfur and the current density was in 32–36 mA $cm^{-2}$ by changing the metal cation from $Na^+$ to $Cs^+$ (Fig. 4b, c). We speculate that the smaller ionic hydration number and radius of hydrated cation of $Cs^+$ ($H_2O)_n$ ($n = 6$ for $Cs^+$ versus $n = 7$ for $K^+$ and 13 for $Na^+$)[45,46] result in stronger interactions with $S^{2-}$ on In surfaces and thus higher ability to activate $H_2O$. These results provide further evidence for our hypothesis that sulfur on indium surfaces functions for $H_2O$ activation via interaction with hydrated metal cations in

the double layer (Fig. 3e). Moreover, the use of $Cs^+$ instead of $K^+$ can further enhance the $CO_2RR$ performance of the S2−In catalyst. A formation rate of formate of 1449 $\mu$mol $h^{-1}$ $cm^{-2}$ with a formate FE of 93% could be achieved at −0.98 V versus RHE, significantly better than those reported to date (Supplementary Table 1).

Besides indium, we found that the strategy to enhance the $CO_2RR$ to formate by doping sulfur can be extended to other metals such as bismuth and tin. The doping of sulfur onto Bi and Sn surfaces with a proper amount significantly promoted the formation rate of formate (Fig. 4d). The formation rates of formate reached 767 and 640 $\mu$mol $h^{-1}$ $cm^{-2}$ over S1−Bi and S1 −Sn catalysts at −0.98 V versus RHE, which were 1.4 and 1.5 times higher that of Bi and Sn catalysts without sulfur, respectively. The FE of formate kept almost unchanged or slightly increased at the same time (Supplementary Fig. 24b). These results confirm the generality of our strategy to accelerate the $CO_2RR$ to formate by enhancing $H_2O$ activation through modifying metal surfaces with an appropriate amount of chalcogenide species.

## Discussion

We discovered a powerful sulfur-doped indium catalyst for the electrocatalytic reduction of $CO_2$ to formate with high selectivity at high current density. Over the S2−In catalyst with a sulfur content of 4.9 mol%, high FE of formate (>85%) could be kept at a large range of current density (25–100 mA $cm^{-2}$) for electrocatalytic $CO_2RR$ in aqueous alkaline media. The formation rate of formate reaches 1002 $\mu$mol $h^{-1}$ $cm^{-2}$ with 93% FE at a potential of −0.98 V versus RHE in $KHCO_3$ aqueous solution. The catalyst is also stable. In $CsHCO_3$ aqueous solution, the formation rate of formate increases to 1449 $\mu$mol $h^{-1}$ $cm^{-2}$ with current density of 84 mA $cm^{-2}$ and FE of 93%. To the best of our knowledge, this is the highest formation rate of formate reported to date for the electrocatalytic $CO_2RR$.

Our studies revealed a unique functioning mechanism of surface sulfur species. Instead of directly activating $CO_2$, the presence of sulfur on indium surfaces enhances $CO_2RR$ to formate by accelerating the activation of $H_2O$. We propose that the adsorbed $S^{2-}$ species on indium surfaces can interact with the hydrated metal cations (such as $K^+$ or $Cs^+$) in the double layer, contributing to the dissociation of $H_2O$ to form adsorbed H* species. The adsorbed H* species is responsible for the formation of HCOO* intermediate, the precursor of formate, on indium surfaces. Platinum can also accelerate $H_2$ evolution, but it mainly enhances the adsorption and recombination of H* species, thus decreasing the FE of formate. Selenium and tellurium also promote the $CO_2RR$ to formate without accelerating $H_2$ evolution, but their roles are relatively weaker because of the weaker interaction with hydrated metal cations. The variation of metal cations in the alkaline media also changes the performance and the highest reaction rate has been achieved using $Cs^+$ in aqueous solution. The promoting effect of chalcogen species can be extended to other metal catalysts such as Bi and Sn. The present work offers a simple and useful strategy for designing highly efficient electrocatalyst for $CO_2$ reduction.

## Methods

**Chemicals and materials**. Indium trichloride tetrahydrate ($InCl_3$·$4H_2O$), tin tetrachloride ($SnCl_4$), deuterium water ($D_2O$), selenium powder (Se) and telluric acid dihydrate ($H_2TeO_4$·$2H_2O$) were purchased from Energy Chemical Co. Bismuth trichloride ($BiCl_3$), sodium sulfide nonahydrate ($Na_2S$·$9H_2O$), potassium bicarbonate ($KHCO_3$), thioacetamide, N,N-dimethylformamide (DMF), dimethylsulfoxide (DMSO) and acetone were purchased from Sinopharm Chemical Co. In foil (99.99%, 0.25 mm thickness) and platinum (II) 2,4-pentanedionate ($Pt(acac)_2$) were purchased from Alfa Aesar Co. Toray TGP-H-060 carbon paper with fiber morphology was purchased from Fuel Cell Store Co. The ultrapure water used in

all experiments with a resistivity of 18.2 MΩ was prepared using an ultrapure water system.

**Fabrication of S−In catalysts**. Carbon fibers were cleaned in acetone and water by sonication for 30 min. $InCl_3$·$4H_2O$ (0.40 mmol) was added into DMF (15 mL), and then 0, 6, 16, 33, or 53 $\mu$mol thioacetamide was added. After vigorous stirring for 15 min, the mixture was transferred into a Teflon-lined autoclave (25 mL). Subsequently, a piece of carbon fibers (3 cm × 1 cm) was tilted in the autoclave. The autoclave was then sealed and heated at 150 °C for 12 h. After cooling to room temperature, the carbon paper was taken out by a tweezer, mildly sonicated in water for 30 s, rinsed with water thoroughly and then dried in an oven at 60 °C overnight. Finally, the electrode was reduced in $CO_2$-saturated 0.5 M $KHCO_3$ aqueous solution at –0.98 V (versus RHE) for 5 min. The mass loading of catalyst on carbon paper was also quantified. For this purpose, the weight difference of carbon fibers before solvothermal treatment and after electroreduction was measured by a micro balance. The above S−In catalysts had the same mass loading of 0.5 ± 0.1 mg $cm^{-2}$. With increasing the feeding amount of thioacetamide from 0 to 53 $\mu$mol during the solvothermal treatment, the catalysts with different sulfur contents denoted as S0−In, S1−In, S2−In, S3−In, and S4−In were obtained.

**Fabrication of S-impregnated S0−In and In foil catalysts**. The S0−In catalyst, which was fabricated above and did not contain sulfur, was also modified with sulfur by an impregnation method. The S0−In catalyst was impregnated in $Na_2S$ aqueous solutions with different concentrations (0.10, 0.25, 0.50, and 1.0 mM) for 5 min Then, the catalyst was dried and was used for the $CO_2RR$. The obtained samples were denoted as S-impregnated S0−In and the sulfur contents measured by Auger electron spectroscopy were 1.4, 2.6, 5.2, and 7.1 mol%. The S-impregnated In foil samples were prepared by the same procedure. In foil was first etched in 5.0 M HCl for 5 min to remove native oxides or impurities under the protection of $N_2$ atmosphere. The pretreated In foil was then impregnated in $Na_2S$ aqueous solutions with concentrations of 0.10, 0.25, 0.50, and 1.0 mM for 3 min under the protection of $N_2$, obtaining S-impregnated In foil samples with sulfur contents of 0.8, 2.2, 4.0, and 7.0 mol%, respectively.

**Fabrication of Se−In and Te−In and Pt−In catalysts**. The procedures for preparation of these modified indium catalysts were the same as those for the preparation of the S−In catalysts, except for using selenium powder, $H_2TeO_4$·$2H_2O$ and $Pt(acac)_2$ to replace thioacetamide.

**Fabrication of S−Sn and S−Bi catalysts**. $SnCl_4$ (20 $\mu$L) was added into DMF (12 mL), and then 0, 6, 12 or 24 $\mu$mol thioacetamide (corresponding to S0−Sn, S1−Sn, S2−Sn, S3−Sn) was added. After vigorous stirring for 15 min, the mixture was transferred into a Teflon-lined autoclave (15 mL). The autoclave was then sealed and heated at 180 °C for 24 h. After cooling to room temperature, the products were collected and washed with ethanol and water, and then dried in an oven at 60 °C overnight. To prepare the electrode, 10 mg sample and 20 $\mu$L Nafion solution (5 wt%) were dispersed in 1.0 mL of isopropanol-water solution with a volume ratio of 3:1 by sonicating for 2 h to form a homogeneous ink. Then, 50 $\mu$L of the suspension was loaded onto a 1 cm × 1 cm carbon paper. The procedures for fabrication of S−Bi catalysts were the same as those for the fabrication of S−Sn, except for using 0.2 mmol $BiCl_3$ instead of 20 $\mu$L $SnCl_4$.

**Characterization**. Powder X-ray diffraction (XRD) patterns were recorded on a Rigaku Ultima IV diffractometer using Cu $K_\alpha$ radiation (40 kV, 30 mA). Scanning electron microscopy (SEM) measurements were performed on a Hitachi S-4800 operated at 15 kV. High-resolution transmission electron microscopy (HRTEM) and energy-dispersive X-ray spectroscopy (EDX) measurements were carried out on a Phillips Analytical FEI Tecnai 20 electron microscope operated at an acceleration voltage of 200 kV. The Auger electron spectroscopy measurements were performed on a PHI 660 operated at 5 kV. The X-ray absorption near edge structure (XANES) spectroscopic measurements were carried out at the BL14W1 beamline of the Shanghai Synchrotron Radiation Facility (SSRF). The X-ray photoelectron spectroscopy (XPS) measurements were performed on a Quantum 100 Scanning ESCA Microprobe (Physical Electronics) using Al $K_\alpha$ radiation (1846.6 eV) as the X-ray source. The $CO_2$ adsorption isotherms were measured at 35 °C with ASAP2020C Micromeritics apparatus.

**Electrochemical measurements**. All electrochemical measurements were carried out on a CHI 760e electrochemical workstation in a three-electrode configuration cell using as-prepared electrode as the working electrode, platinum plate (2 cm × 2 cm) as the counter electrode, and SCE as the reference electrode in 0.5 M $KHCO_3$ aqueous electrolyte (pH = 7.2). The electrode area was controlled at 1 cm × 1 cm. Anode and cathode compartments contained 30 mL electrolyte with gas headspace of 20 mL, and were separated by a proton exchange membrane (Nafion-117). The scheme of the electrocatalysis setup is shown in Supplementary Fig. 25a. Before the measurement, the working electrode (cathode) compartment was purged with $CO_2$ with a flow rate of 20 mL $min^{-1}$ for 30 min to obtain $CO_2$-saturated electrolyte. During electrocatalytic reactions, the solution in cathode

compartment was vigorously stirred at a speed of 2000 rpm using a magnetic stirrer. The effluent gas from the cathode compartment went through the sampling loop of gas chromatograph and was analyzed on line. $H_2$ was analyzed by thermal conductivity detector (TCD). CO and hydrocarbons were analyzed by flame ionization detector (FID). Liquid products were analyzed by $^1H$ NMR spectroscopy. The $^1H$ NMR spectrum was recorded on an Advance III 500-MHz Unity plus spectrometer (Bruker), in which 0.5 mL of the electrolyte was mixed with 0.1 mL DMSO (internal standard, diluted to 100 ppm (v/v) by deuterated water) (Supplementary Fig. 25b). The HER performance was measured in 1.0 M KOH electrolyte under continuous purging with $N_2$ gas, and Hg/HgO (1.0 M KOH) electrode was used as the reference electrode.

Electrocatalytic $CO_2RR$ in $D_2O$ solution was performed with similar procedures except for replacing $H_2O$ with $D_2O$. For product analysis, the total amount of $HCOO^-$ and $DCOO^-$ in aqueous phase was quantified by HPLC after the reaction. The amount of $HCOO^-$ produced was determined by $^1H$ NMR. Then, the amount of $DCOO^-$ was calculated by subtracting the amount of $HCOO^-$ from the total amount of $HCOO^-$ and $DCOO^-$.

For gaseous products, the Faradaic efficiency was calculated as follows. The molar flow of gas from the electrochemical cell was calculated using the concentration of species $g$ measured by GC ($x_g$ (mol mol$^{-1}$)) and the $CO_2$ flow rate ($F_{CO_2}$ (mol s$^{-1}$)). With the number of exchanged electrons to produce species $g$ from $CO_2$ ($n_g$) and Faraday constant (96,485 C mol$^{-1}$), the partial current towards species $g$ ($i_g$ (A)) was calculated. Comparing the partial current to the total current ($i_{tot}$ (A)) yielded the Faradaic efficiency for species $g$ (FE$_g$):

$$FE_g = i_g/i_{tot} = 96,485 \times x_g \times F_{CO_2} \times n_g/i_{tot} \qquad (3)$$

For liquid products, the following method was used for the calculation of Faradaic efficiency. The concentration of formate $c_l$ (mol L$^{-1}$) was calculated from the standard curve shown in Supplementary Fig. 25c. With $n_l$, Faraday constant and the electrolyte volume in the cell ($V_{cell}$ (L)), the partial charge to produce species $l$ ($q_l$ (C)) was calculated. Comparing the partial current to the total charge passed ($q_{tot}$ (C)) yielded the Faradaic efficiency for species $l$ (FE$_l$):

$$FE_l = q_l/q_{tot} = 96,485 \times c_l \times V_{cell} \times n_l/q_{tot} \qquad (4)$$

The formation rate for all species were calculated using the following equation:

$$\text{Formation rate} = (q_{tot} \times FE)/(96,485 \times n \times t \times S) \qquad (5)$$

where $t$ was the electrolysis time (h) and $S$ was the geometric area of the electrode (cm$^2$). In all $CO_2RR$ measurements, we used SCE as the reference. It was calibrated with respect to RHE: $E$ (RHE) = $E$ (SCE) + 0.241 + pH × 0.0592. All the electrocatalytic reactions were conducted at room temperature, and 85% IR correction was applied in all the measurements.

We evaluated the current density of $CO_2RR$ under mass-transport limitation ($j_{limit}$) using the following equation:

$$j_{limit} = n \times F \times D \times C/\delta \qquad (6)$$

Here, $n$ represents the number of electrons per $CO_2$ reacted, which is 2 here because formate and CO are the dominant products. $F$ is the Faraday constant ($F$ = 96485 C mol$^{-1}$). $D$ is the diffusion coefficient of $CO_2$ (2.02 × 10$^{-9}$ m$^2$ s$^{-1}$). $C$ represents the saturated bulk concentration of $CO_2$, which is 34 mol m$^{-3}$ at 1 bar and 25 °C. $\delta$ is the diffusion layer thickness for $CO_2$, which can be estimated from rotating disk electrode model with the Levich equation:

$$\delta = 1.61 \times D^{1/3} \times v^{1/6}/\omega^{1/2} \qquad (7)$$

where $v$ is kinematic viscosity of electrolyte (1.0 × 10$^{-6}$ m$^2$ s$^{-1}$) and $\omega$ is the angular frequency of rotation, which is 2 π × rotation rate (s$^{-1}$). To make an accurate evaluation of the diffusion layer thickness, we performed linear sweep voltammetry (LSV) measurements using the rotating disk electrode and the magnetic-stirrer agitation. See Supplementary Methods and Supplementary Fig. 26 for details.

**Density functional theory (DFT) calculations**. DFT calculations were performed by using the Perdew-Burke-Ernzerhof (PBE) functional of generalized gradient approximation (GGA)[47] in Vienna Ab Initio Simulation Package (VASP)[48]. The projector-augmented wave (PAW) method was applied to describe the electron-ion interactions[49]. Energy cutoff of 450 eV was used, and the Methfessel-Paxton method with a broadening factor is 0.1 eV. In this work, the vacuum space of 15 Å was used for all calculations. Γ-centered k-point sampling grid of 6 × 6 × 1 was adopted. The (101) surface with 2 × 2 six-layer slab were relaxed for the top four layers. On the other hand, because surface formate species takes a unit negative charge, the present DFT calculation is not so good as to describe this kind of system carrying neat charge. Thus, HCOOH was considered as the final product to describe this reaction instead of formate, in line with the DFT calculation practice in many studies[20,37,50].

In this work, all thermodynamic properties were further calculated using the Atomic Simulation Environment suite of programs[51]. The Gibbs free energies were

calculated at 25 °C and 1 atm,

$$G = H - TS = E_{DFT} + E_{ZPE} + \int_0^{298\,K} C_v dT - TS \qquad (8)$$

where $E_{DFT}$ is the total electronic energy obtained from DFT optimization, $E_{ZPE}$ is the zero-point vibrational energy (ZPE) obtained from calculated vibrational frequencies, the thermal energy $\int_0^{298\,K} C_v dT$ is calculated from the heat capacity, $T$ is the temperature, and $S$ is the entropy. The ideal gas approximation and the harmonic approximation were used for $CO_2$, $H_2$, CO, HCOOH, and $H_2O$ molecules, and for adsorbates all atomic nuclear motions were considered as harmonic oscillators.

## Data availability
The data that support the findings of this study are available from the corresponding authors upon a reasonable request.

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

## Acknowledgements

This work was supported by the National Key Research and Development Program of the Ministry of Science and Technology of China (No. 2017YFB0602201), the National Natural Science Foundation of China (Nos. 21690082, 91545203, and 21503176). We thank staff at the BL14W1 beamline of the Shanghai Synchrotron Radiation Facilities (SSRF) for assistance with the EXAFS measurements.

## Author contributions

W.M. and S.X. performed most of the experiments and analyzed the experimental data. X.Z. performed computational studies and analyzed the computational data. F.S and Z.J. conducted XANES measurements and analyzed the results. J.K. conducted a part of characterizations. Q.Z. analyzed all the data and co-wrote the paper. D.W. guided the computational work, analyzed all the data and co-wrote the paper. Y.W. designed and guided the study, and co-wrote the paper. All of the authors discussed the results and reviewed the manuscript.

## Additional information

**Competing interests:** The authors declare no competing interests.

