## [Peer Review File · Nature Communications]

Reviewers' comments:

Reviewer #1 (Remarks to the Author):

Wang and co-workers report an enhancement effect of sulfur on electrochemical CO₂ reduction to formate catalysed by indium. The study is systematic and the manuscript is also well prepared. The results obtained by the authors are important and of broad interest to the readers of Nature Communications. However, I found that the current density reported by the authors is amazingly large (i.e. ~100 mA cm⁻²), which may be well above the mass transport limited value. The authors should give an estimate about the mass transport limited current. Without this information, I cannot comment on the suitability of this paper for publication.

Reviewer #2 (Remarks to the Author):

In this manuscript, authors report nanoparticulated indium modified with sulfur as novel catalysts for the electrocatalytic reduction of CO₂ to formate in aqueous media. They screen different amounts of sulfur contents and determine the optimum for high activity. Subsequently, they expand the strategy to other materials.

General comments:

I have read with interest this manuscript, as the modification of matrices with chalcogens start to show unexpected performance for this reaction. The approach to discover the material bearing in mind its final application is a positive aspect and it shows potential for scalability. Performance is remarkable though not extraordinary mainly due to high overpotentials, with room for improvement. However, I cannot recommend publication in its current state due to some incoherent/not justified statements and, critically, to the lack of novelty/insight the demanding scientific level of Nat. Commun. requires. In more detail, various concepts are entangled in the manuscript:

- Authors do not mention the similar effects encountered over Cu when modified with sulfur, which transforms a non-selective catalyst toward HCOO⁻ such as Cu into a highly selective one, in contrast to naturally selective In (ACS Catal. 2018, 8, 837–844., ChemSusChem, 2017, 11, 320–326), thus making this strategy not novel.
- Authors strongly point to the activity of the electrodes as the key scientific finding. However, current density strongly depends on non-catalytic parameters such as dispersion, loading, porosity,....and thus is associated, but not fully representative, of the inherent characteristics of the S-In systems. Authors must clearly separate inherent activity of S-In from the activity of the prepared electrodes. In addition, the field is not yet in a race to obtain large current densities, given its early stage of development. Lastly, applied overpotentials are very large to be considered as practical, which reduces its impact. I thus consider the otherwise promising current densities reported of high, but not utmost interest.
- From data, it seems clear that the high selectivity associated to bulk indium is improved from the nanostructuring (see S0-In vs. foil), whereas the role of sulfur is to improve activity. I consider the development of these facts the main messages to provide aiming rational design principles (see below specific comments).

Specific comments:

- The comparison in Fig. 1b is unfair, since the ECSA is not considered.
- I recommend Suppl. Fig. 2 to be transferred to the main text with the ECSA correction implemented. Please keep the use of bars and dots consistent in the supplementary data and manuscript.
- A parallel study on the modification of an In foil with sulfur would help disentangle the nanostructuring effect and the porosity of the electrode from the effect sulfur brings to In surfaces
- In spite of the consistent evidence of In³⁺ and In⁰ in the catalyst even under reaction conditions, authors do not comment further. There is reported evidence of the role of In hydroxide species in the electrocatalytic reduction of CO₂ (ref. 21, used in the text to assign XPS peaks, or ACS Catal. 2016, 6, 6265–6274.)
- Though reasonable, the mechanistic reason aiming to explain the role of sulfur is loosely bound to the DFT calculations. Authors observe a large predominance of the (101) facet in XRD and also in TEM analysis, in contrast to the (101) selected for the theoretical study. Authors must justify this selection. In parallel, authors claim water acting as the proton donor as the key step, but DFT calculations leave this step aside as they assume H⁺ in solution. Though aware of the limitations of DFT, authors should find a more solid background for their hypothesis to strengthen their conclusions.
- The shown enhancement in the formate production rate upon modification with Pt is within the error bars and, as authors acknowledge, was not a successful attempt. I suggest to move it to the supplementary information.

Reviewer #3 (Remarks to the Author):

This is a rather interesting paper that reports for the first time that the presence of sulfide on the surface of an indium (or other heavy p-group metal) electrode enhances the electrodes electrocatalytic activity toward CO₂ reduction to formate in a basic aqueous electrolyte. The authors report a high faradaic efficiency that is maintained even at high current densities – a finding the authors claim is unique. In fact, though rare, other indium-based systems will do this also, and are of industrial interest. (See for example: Journal of CO₂ Utilization 7, (2014) p1–5) Nonetheless, the authors finding that sulfur treated electrodes behavior in this manner is intriguing and novel from the chemical point of view. This finding leads the authors to suggest that the role of the surface sulfide is to activate the formation of a surface hydrogen atom formed from a water ligated to a supporting electrolyte cation. To probe this possibility, the researchers cleverly exam the effect of varying the alkali cation present in the electrolyte on the formation of formate. They report that as one drops down the first column of the periodic table (reducing the number of waters ligated to the cation) that the catalytic efficiency improves, and argue that a smaller hydrated cation will interact more strongly with the surface sulfide anion. The papers conclusions are supported by an ample set of experimental data covering issues of surface science, electrochemistry and quantum chemistry simulation.

Yet, given the extensive experiments and thought that has gone into this paper there are some surprising omissions:

- A ¹³C₂O₂ experiment is not reported to demonstrate that CO₂ is in fact the source of the observed products. This is considered a standard control in the area of CO₂ electrochemistry at this point.

- A pH dependence is not undertaken, even though the author's mechanism requires a basic electrolyte and a pH study would certainly shed more light on the mechanism of CO₂ reduction.
- **MOST IMPORTANTLY:** Though the authors cite at least one paper (reference 20) that argues that surface oxides are responsible for the electrocatalytic activity of In with regard to CO₂, the authors never mention the existence of a surface oxide or what its role might be in their system. Given, both the electrochemistry and surface spectroscopy carried out by the authors, it would be impossible to miss the existence of a surface oxide. Thus, this is either a serious omission, i.e. they elected to not report the presence of surface oxides, or a chemical miracle – adding a submonolayer of sulfide to an indium surface protects it from oxide formation. Given the intense reactivity of “normal” indium with air to form an instantaneous oxide coating – I doubt that the sulfur is suppressing this reaction. If it is, then they have a much bigger finding in the area of corrosion chemistry than the CO₂ chemistry they are reporting. Their lack of reporting the nature of the surface oxide therefore throws the whole study into doubt. It also challenges their DFT study, since their model does not consider the presence or role of a surface oxide. Once this point is dealt with honestly, I will enthusiastically support the publication of this work.
- Finally, I note that the experiments involving the addition of sodium sulfide to the electrolyte are not sufficiently clear to reproduce. Is the sulfide present in the CO₂ purged electrolyte or does the sulfide exposure involve a pretreatment of the electrode?

Responses to Reviewers

Response to Reviewer 1

General comments: Wang and co-workers report an enhancement effect of sulfur on electrochemical CO₂ reduction to formate catalysed by indium. The study is systematic and the manuscript is also well prepared. The results obtained by the authors are important and of broad interest to the readers of Nature Communications. However, I found that the current density reported by the authors is amazingly large (i.e. ~100 mA cm⁻²), which may be well above the mass transport limited value. The authors should give an estimate about the mass transport limited current. Without this information, I cannot comment on the suitability of this paper for publication.

Reply and actions taken: We thank the reviewer for the pertinent comments on our work. We have evaluated the current density under the mass transport limitation (j_{limit}) by using the following equation based on the semi-infinite diffusion model proposed by Bard and Faulkner (Bard, A. J. & Faulkner, L. R. *Electrochemical methods: fundamentals and applications*. Wiley 2001):

$$j_{\text{limit}} = n \times F \times D \times c / \delta \quad (1)$$

where n represents the number of electrons per CO₂ reacted, which is 2 here because formate and CO are the dominant products. F is the Faraday constant ($F = 96485 \text{ C mol}^{-1}$). D is the diffusion coefficient of CO₂ ($2.02 \times 10^{-9} \text{ m}^2 \text{ s}^{-1}$). c represents the saturated bulk concentration of CO₂, which is 34 mol m^{-3} at 1 bar and 25 °C. δ is the diffusion layer thickness for CO₂. δ can be roughly estimated using the rotating disk electrode model with the Levich equation:

$$\delta = 1.61 \times D^{1/3} \times \nu^{1/6} / \omega^{1/2} \quad (2)$$

where ν is the kinematic viscosity of electrolyte ($1.0 \times 10^{-6} \text{ m}^2 \text{ s}^{-1}$), and ω is the angular frequency of rotation (209 s^{-1}). Thus, δ was calculated to be 14.0 μm . By using Eq. 1, we have calculated the current density under mass transport limitation (j_{limit}). The value is 94 mA cm⁻². This value of j_{limit} is close to that reported in previous work under similar experimental conditions (Yan, C. et al., *Energy Environ. Sci.* **11**, 1204-1210 (2018)).

In our work, the current density is composed of two parts, i.e., the one part contributed by CO₂RR and the other part by HER. Thus, at a current density of 100 mA cm⁻², the current density ascribed to CO₂RR is 86 mA cm⁻² by considering that the Faradic efficiencies (Fes) of formate and CO are 85% and 1%, respectively, while the FE of H₂ is 14%. This value of $j(\text{CO}_2\text{RR})$ (86 mA cm⁻²) approaches the j_{limit} (94 mA cm⁻²). It should be noted that two recent papers have reported $j(\text{CO}_2\text{RR})$ values close to our result under similar reaction conditions for the reduction of CO₂ to CO.

An Au nano-needle catalyst exhibited a current density of $\sim 80 \text{ mA cm}^{-2}$ for the reduction of CO_2 to CO (Liu, M. et al., *Nature* **537**, 382-386 (2016)). A coordinatively unsaturated Ni-N catalyst exhibited a current density of 73 mA cm^{-2} for the reduction of CO_2 to CO (Yan, C. et al., *Energy Environ. Sci.* **11**, 1204-1210 (2018)). On the other hand, the current density for the reduction of CO_2 to formate reported to date is $< 55 \text{ mA cm}^{-2}$ (Supplementary Table 1). These facts indicate that the current density of 85 mA cm^{-2} for CO_2RR to formate observed in our work is reasonable and also suggest the superior performance of our catalyst for CO_2RR to formate.

It is noteworthy that our mechanistic studies on the effect of sulfur were carried out under the conditions, where the current density of CO_2RR was much lower than the j_{limit} . This means that the reaction rate or catalytic activity is not limited by mass transportation. To make this point clear, we have calculated the current density of CO_2 reduction ($j(\text{CO}_2\text{RR})$) for the S2-In catalyst by using the total current density and the FE of CO_2RR products (formate and CO) at different applied potentials, and the result has been added in Supplementary Fig. 4 in the revised manuscript. The result reveals that the $j(\text{CO}_2\text{RR})$ increases almost exponentially from -0.33 to -1.03 V versus RHE, demonstrating that the activity is controlled by the reaction on the electrocatalyst in this region. The increase in $j(\text{CO}_2\text{RR})$ slows down significantly when the potential is more negative than -1.13 V, and $j(\text{CO}_2\text{RR})$ becomes saturated at $> 80 \text{ mA cm}^{-2}$ at -1.23 V versus RHE. This observation suggests that the activity is determined by the mass transportation at potentials more negative than -1.13 V versus RHE. Our studies were mostly conducted at -0.98 V versus RHE with current density of CO_2RR significantly lower than 80 mA cm^{-2} . Thus, in most cases, the electrocatalytic activity is not determined by the mass transportation.

We have added the following sentences in the revised manuscript to explain the results and discussion described above: “*The current density ascribed to CO_2RR , which was calculated by considering the FE of CO_2RR , increased significantly from 0.030 to 86 mA cm^{-2} by changing potential from -0.33 to -1.23 V versus RHE and then became saturated (Supplementary Fig. 4). It is noteworthy that the current density of CO_2RR of 86 mA cm^{-2} approaches the maximum value (94 mA cm^{-2}) evaluated by assuming the mass-transport limitation under our reaction conditions (see Supplementary Methods)” (please see Page 6, Lines 6-12).*

The procedure for the evaluation of the current density under mass-transport limitation described above has been added in the Supplementary Methods (please see Supplementary Information, Page 21).

Response to Reviewer 2

General Comments: In this manuscript, authors report nanoparticulated indium modified with sulfur as novel catalysts for the electrocatalytic reduction of CO₂ to formate in aqueous media. They screen different amounts of sulfur contents and determine the optimum for high activity. Subsequently, they expand the strategy to other materials. I have read with interest this manuscript, as the modification of matrices with chalcogens start to show unexpected performance for this reaction. The approach to discover the material bearing in mind its final application is a positive aspect and it shows potential for scalability. Performance is remarkable though not extraordinary mainly due to high overpotentials, with room for improvement. However, I cannot recommend publication in its current state due to some incoherent/not justified statements and, critically, to the lack of novelty/insight the demanding scientific level of Nat. Commun requires. In more detail, various concepts are entangled in the manuscript.

Reply and actions taken: We thank the reviewer for the kind evaluation of our manuscript. We also appreciate the critical comments by this reviewer to improve the quality of our manuscript. We have performed additional studies to disentangle various factors, provide clear insights and answer the questions and comments raised by the reviewer. The major novelty and new insights of our work are highlighted as follows.

First, the S–In catalyst reported in this work exhibits significantly high activity and selectivity for CO₂ reduction to formate than the catalysts reported to date (Supplementary Table 1). In particular, the high FE of formate (> 85%) can be maintained in a large range of current density (25-100 mA cm⁻²) over the S2–In catalyst, whereas the FE of formate usually drops significantly to < 60% at a current density of > 60 mA cm⁻² over the electrocatalysts reported to date. Thus, the formation rate of formate that we have achieved significantly breaks the current upper limit of 1000 μmol g⁻¹ h⁻¹, reaching 1449 μmol g⁻¹ h⁻¹ with an FE of formate of 93% at an applied voltage of –0.98 V versus RHE.

Second, the reviewer has made criticism on the high overpotentials. Although to achieve high activity (current density) and selectivity (FE of formate) at a lower overpotential is ideal, the keeping of high FE of formate (>85%) over our S2–In catalyst in a large range of current density as well as the highest formation rate of formate reported in this work also represents a significant step forward in the field of electrocatalytic reduction of CO₂. We have also added information of catalytic behaviors of our catalysts at relatively lower overpotentials. The S2–In catalyst also

shows better performances at low overpotentials than most of the non-noble metal catalysts, in particular the p-block element-based catalysts (please see Table R1). For example, our S2–In catalyst provides an FE of formate of 80% at an overpotential of 0.44 V (applied potential of –0.63 V) with a formation rate of formate of 75 $\mu\text{mol h}^{-1} \text{cm}^{-2}$, whereas the FEs were lower than 65% for Sb nanosheets, porous SnO₂ and S-modified Cu catalysts. These results at lower overpotentials have also been added in the Supplementary Table 1.

Table R1. Some typical results of non-noble metal catalysts for photocatalytic reduction of CO₂ to formate at low overpotentials.

Catalyst	Overpotential ^a (V)	Current density (mA cm ⁻²)	FE of formate (%)	Formation rate of formate ($\mu\text{mol h}^{-1} \text{cm}^{-2}$)	Ref.
Sb nanosheets	0.47	2	50	19	Angew. Chem. Int. Ed. 56 , 14718-14722 (2017)
Porous SnO ₂	0.48	6	35	39	Angew. Chem. Int. Ed. 56 , 505-509 (2017)
S-modified Cu	0.41	2.5	50	23	ACS Catal. 8 , 837-844 (2018)
S-Cu ₂ O derived Cu	0.46	5.3	64	63	ChemSusChem , 11 , 320-326 (2018)
S-In	0.44	5.0	81	75	This work

^a Overpotential is calculated using the equation of –0.19 V (theoretical minimum voltage for CO₂ reduction to formate) – applied potential (V).

Third, our work has demonstrated a new functioning mechanism of sulfur species on In surfaces. Sulfur accelerates the activation of H₂O, forming active hydrogen species that can readily react with CO₂ to produce formate. This is quite different from the current consensus that the enhancement in the activation of H₂O would accelerate H₂ formation, the competitive reaction with the CO₂RR, and thus decrease the FE of CO₂RR. This enables the increase in the current density or activity while keeping a high FE of CO₂RR. Furthermore, we have demonstrated that such an effect can be extended to other chalcogen promotes and to other p-block metal catalysts, and thus the work provides a universal and simple method for designing electrocatalysts with both high activity and selectivity for CO₂ reduction. In short, the present work can not only shed light on the importance of water activation in CO₂ reduction but also offers a methodology to gain high FE of CO₂RR (formate) at high reaction rate

(high current).

General Comment 1: Authors do not mention the similar effects encountered over Cu when modified with sulfur, which transforms a non-selective catalyst toward HCOO^- such as Cu into a highly selective one, in contrast to naturally selective In (*ACS Catal.* 2018, **8**, 837–844, *ChemSusChem*, 2017, **11**, 320–326), thus making this strategy not novel.

Reply and actions taken: We have read these two references with interests and cited them in our revised manuscript as Refs. 21 and 22. The related results reported in these references have been added in Supplementary Table 1. As pointed out by the reviewer, these two articles reported that the non-selective Cu catalysts became selective for the formation of formate after the modification by sulfur. The results are, of course, interesting. However, we cannot agree with the reviewer that these articles would decrease the novelty and the impact of our work because of the following reasons.

First, the S–In catalyst reported in our work, which represents chalcogen-modified p-block element catalysts, is a different type of catalysts from the modified Cu catalysts. In our work, the nanosized In catalyst itself already exhibits relatively high FE of formate, and the sulfur modification significantly accelerates the activity for formate formation while keeping the high FE of formate. This enables us to attain high activity, i.e., current density, at high selectivity, i.e., FE of formate. We believe that this is a very important step forward, since the FE of formate during CO_2RR usually drops at a high current density ($> 60 \text{ mA cm}^{-2}$) over the electrocatalysts reported to date due to the enhancement in HER, although high values of FE of formate have been reported in many papers. Thus, a high formation rate of formate can be obtained over the S–In catalyst. Our S₂–In catalyst demonstrates a formation rate of formate of $1449 \mu\text{mol h}^{-1} \text{ cm}^{-2}$ with 93% FE of formate. This rate is 6.6–10 times higher than that over the sulfur-modified Cu catalyst, which is $146\text{--}220 \mu\text{mol h}^{-1} \text{ cm}^{-2}$ with 80% FE of formate (Supplementary Table 1; *ACS Catal.* **8**, 837–844 (2018), *ChemSusChem* **11**, 320–326 (2018)). In addition, high FEs of formate ($>80\%$) can be achieved over the S–In catalyst in a wide range of overpotentials (0.44–1.24 V) with current densities varied from 5 to 100 mA cm^{-2} . On the other hand, over the sulfur-modified Cu catalyst, $\sim 80\%$ FE of formate can only be obtained at an overpotential of 0.8 V with a current density of $10 \pm 5 \text{ mA cm}^{-2}$ (*ACS Catal.* **8**, 837–844 (2018), *ChemSusChem* **11**, 320–326 (2018)). These results clearly demonstrate the unique feature of the present S–In catalyst.

Second, as reported in the two references, it is proposed that the sulfur-induced change in the major products (from CO or C_2H_4 as well as other CO_2 -reduction

products to formate) over Cu catalysts is mainly due to the reconstruction of catalysts by sulfur, and such reconstruction alters the particle size and surface state of Cu (*ACS Catal.* **8**, 837–844 (2018)). The presence of sulfur on Cu is speculated to disfavor the adsorption of the *COOH intermediate, and thus is unbeneficial to the formations of CO or CO-derived products, which is a competitive reaction of formate formation. On the other hand, in the present work, we have demonstrated a completely different functioning mechanism of sulfur. We have confirmed that the nanostructure (SEM, HRTEM), the chemical state (XPS, EXAFS), and the electrochemical surface areas (ECSA) of In catalysts do not undergo significant changes after sulfur modification for the S–In series of catalysts. The amount of CO₂ adsorption on catalyst surfaces also did not change significantly after sulfur modification. Our experimental studies and DFT calculations both reveal that the sulfur modification boosts the activation of H₂O on In surfaces, forming active hydrogen species, which can easily react with CO₂ to formate. Our work shed light on the importance of activation of H₂O in CO₂RR, which has been overlooked in previous studies.

Furthermore, we have demonstrated that the effect of sulfur on In surfaces can be extended to other p-block element catalysts such as Bi and Sn. Other chalcogen elements such as Se and Te also exhibit enhancing effect on formate formation over In catalysts. Therefore, our work offers a simple and useful strategy for designing electrocatalyst for CO₂ reduction with high FE of formate at high reaction rate. This is not the case for the chalcogen modifier for the Cu-based catalysts. As indicated in the reference, Ag–S and Cu–Se catalysts did not exhibit any significant enhancement toward the reduction of CO₂ to formate (*ChemSusChem* **11**, 320–326 (2017)).

In short, although sulfur is used as the promoter in both our work and the papers dealing with Cu catalysts (Refs. 21 and 22), our present work reports a new promoting effect of sulfur on the reduction of CO₂ to formate and indicates a novel functioning mechanism of sulfur. Sulfur on In surfaces significantly increases the activity (current density to ~100 mA cm⁻²) while keeping the high FE of formate (>85%), and thus high formation rate of formate (1449 μmol h⁻¹ cm⁻²) could be attained at FE of formate of 93%. Our work demonstrates that the sulfur modification boosts the activation of water on In surface, forming active hydrogen species, which can easily react with CO₂ to formate. Furthermore, such an enhancing effect can be extended to other p-block element catalysts such as Bi and Sn and other chalcogen elements such as Se and Te.

General Comment 2: Authors strongly point to the activity of the electrodes as the key scientific finding. However, current density strongly depends on non-catalytic parameters such as dispersion, loading, porosity,...and thus is associated, but not

fully representative, of the inherent characteristics of the S-In systems. Authors must clearly separate inherent activity of S-In from the activity of the prepared electrodes. In addition, the field is not yet in a race to obtain large current densities, given its early stage of development. Lastly, applied overpotentials are very large to be considered as practical, which reduces its impact. I thus consider the otherwise promising current densities reported of high, but not utmost interest.

Reply and actions taken: We totally agree with the reviewer that some non-catalytic morphological parameters such as dispersion, loading and porosity of electrocatalysts as well as the electrode may also influence the current density. Actually, during our study on the effect of sulfur modification, we have excluded the influences of these non-catalytic factors. First, we employed the same electrode with the same carbon fiber (Toray TGP-H-060) as the substrate to load the electrocatalyst. The mass loading of S-In powders on carbon fibers is the same, being $0.5 \pm 0.1 \text{ mg cm}^{-2}$, for the catalysts from S0-In to S4-In with different sulfur contents (*please see Page 19, Lines 5-6*). Our characterizations for the series of S-In (from S0-In to S4-In) catalysts using SEM and HRTEM confirmed that the size or the dispersion of In particles did not change significantly with the sulfur content, and the average sizes were 110-130 nm (*Supplementary Figs. 8 and 9*). The S-modified In particles are essentially non-porous in the catalysts. Furthermore, the morphologies of the catalysts did not undergo significant changes after the reaction (*Supplementary Fig. 8f and 9f*). Therefore, we can exclude the influences of the non-chemical parameters, such as loading, dispersion and porosity, on current density or activity during our study on the effect of sulfur modification. To make this point clear, we have added the following sentences in the revised manuscript: “*The catalyst loading on carbon fibers was $0.5 \pm 0.1 \text{ mg cm}^{-2}$ for each catalyst. These suggest that there are no significant differences in non-chemical parameters for the S-In series of catalysts, such as catalyst loading, size or dispersion of In particles and catalyst porosity. Thus, these parameters do not account for the enhanced current density and the formation rate of formate after the modification of In catalysts by sulfur*” (*please see from Page 7-the last line to Page 8-Line 5*).

As pointed out by the reviewer, we also do not think that the current density alone should be over-emphasized. Instead, we hope to emphasize that the keeping of high FE at a high current density is the key to CO₂RR. It is generally accepted that for CO₂RR, the FE, current density and overpotential are all important (*Joule* **2**, 825-832 (2018); *Angew. Chem. Int. Ed.* **56**, 2-30 (2017)). Actually, many previous papers have reported catalysts with high FEs of formate for CO₂RR, but the high FE of formate can only be sustained in a limited range of current density and it drops at a high current density (*Supplementary Table S1*). Thus, the formation rate of formate

reported to date is limited to $1000 \mu\text{mol h}^{-1} \text{cm}^{-2}$. In this work, we found that high FE of formate (>85%) could be attained in a broad range of current density (5-100 mA cm^{-2}), and the formation rate of formate reached $1449 \mu\text{mol h}^{-1} \text{cm}^{-2}$ at FE of formate of 93%. We have demonstrated that this arises from the effect of sulfur on In surfaces, which can significantly increase the activity (current density to $\sim 100 \text{mA cm}^{-2}$) while keeping the high FE of formate (>85%). We believe that this is a significant step forward in designing efficient electrocatalysts for CO_2RR with both high activity and selectivity.

We agree with the reviewer that the overpotential is also an important factor. Thus, we have added the information of behaviors of S-In catalysts at relatively lower overpotentials in the revised manuscript. As displayed in Supplementary Fig. 3, the CO_2RR started to occur at an overpotential of 0.14 V (-0.33 V versus RHE) with 3% FE of formate over the S2-In catalyst, and this is comparable to that achieved with partially oxidized Co (~ 0.10 V with 2.3% FE of formate) (*Nature* **529**, 68-71 (2016)). Typically, to reach a high FE of formate ($\geq 80\%$), an overpotential of at least 0.8 V is required when a p-block element is used as a catalyst (*ACS Catal.* **8**, 837-844 (2018)). As mentioned above, the S2-In catalyst also shows better performances at low overpotentials than most of the non-noble metal catalysts, in particular the p-block element-based catalysts (please see Table R1). For examples, our S2-In catalyst provides a FE of formate of 80% at an overpotential of 0.44 V (potential of -0.63 V versus RHE) with a formation rate of formate of $75 \mu\text{mol h}^{-1} \text{cm}^{-2}$, whereas the FEs were lower than 65% for Sb nanosheets, porous SnO_2 and S-modified Cu catalysts. These results at lower overpotentials have also been added in Supplementary Table 1 in the revised manuscript. The catalytic behaviors of the S2-In catalyst at potentials ranging from -0.33 to -1.23 V (versus RHE) have been displayed in Supplementary Fig. 3. The results obtained using In foil have also been shown in Supplementary Fig. 3 for comparison. We have added the following sentences to describe the results obtained at lower overpotentials: as “*We performed further studies for the most efficient S2-In catalyst at different cathodic potentials. The CO_2RR started to occur at a potential of -0.33 V versus RHE (overpotential, 0.14 V) with FE of formate of 3% (Supplementary Fig. 3a), comparable to that over a partially oxidized Co catalyst¹¹. 80% FE of formate was achieved at -0.63 V versus RHE (overpotential, 0.44 V), better than those over most of the non-noble catalysts under such a lower overpotential (Supplementary Table 1)” (please see from Page 5-the last paragraph to Page 6-Line 4).*

General Comment 3: From data, it seems clear that the high selectivity associated to bulk indium is improved from the nanostructuring (see S0-In vs. foil), whereas the

role of sulfur is to improve activity. I consider the development of these facts the main messages to provide aiming rational design principles (see below specific comments).

Reply and actions taken: We thank the reviewer for this constructive comment. As pointed out by the reviewer, the S0–In catalyst (without sulfur) fabricated by electroreduction of In_2O_3 precursor growing on carbon fibers exhibits higher selectivity or FE of formate during CO_2RR than In foil. Regarding the activity, our result shows that, although the S0–In catalyst possesses higher formation rate of all products (HCOO^- , CO and H_2) than In foil, the ECSA-corrected formation rates of all products are almost the same for the two catalysts (Supplementary Fig. 16).

Our SEM characterizations show that the S0–In catalyst possesses nanoparticulate morphology with an average diameter of 128 nm (Supplementary Figs. 8a and 9a), while In foil has smooth surfaces (Supplementary Fig. 8g). These results suggest that the nanostructure of In catalysts might be one factor that leads to high selectivity of formate. Furthermore, our EDS and XPS results (Fig. 2b and Supplementary Fig. 13) confirm the presence of a fraction of In^{3+} (In_2O_3) species on the S0–In surface. We speculate that the In^{3+} species and the nanostructure may both contribute to the high FE of formate over the S0–In catalyst.

We have added the following new paragraph to describe the results and discussion mentioned above in the revised manuscript: “*Our present work has demonstrated that the sulfur-modified In catalyst is very promising for electrocatalytic reduction of CO_2 to formate. To understand the role of sulfur more deeply, it is necessary to disentangle different factors that may contribute to CO_2RR in the present system. Our results show that the S0–In catalyst without sulfur fabricated by electroreduction of In_2O_3 precursor growing on carbon fibers exhibits higher FE of formate than In foil (Fig. 1a and Supplementary Fig. 16a). The activity of the S0–In catalyst, expressed by the ECSA-corrected formation rate of all products (including HCOO^- , CO and H_2), is almost the same with that of In foil (Supplementary Fig. 16b). The S0–In catalyst exhibits nanoparticulate morphology with an average diameter of 128 nm, whereas In foil has smooth surfaces (Supplementary Fig. 8). Moreover, our XPS measurements reveal that a small fraction of In^{3+} (i.e., In_2O_3) species co-exists with metallic In on the S0–In surface. The nanostructured morphology and the presence of oxidized species on metal catalysts were reported to be beneficial to CO_2RR ^{23,28-30}. In particular, $\text{In}(\text{OH})_3$ was proposed to play a crucial role in the formation of formate or CO ^{23,30}. Our XPS results indicated the co-existence of In_2O_3 but not $\text{In}(\text{OH})_3$ with In^0 in our case. To understand the role of surface oxidized species, we further pretreated In foil in air at 250 °C for 3 h to generate a coverage of In_2O_3 on In surfaces. The electrocatalytic CO_2RR result showed that the FE of formate increased on the surface-oxidized In foil, although the formation rate of all*

products based on ECSA did not change significantly (Supplementary Fig. 16). Our CO₂-adsorption measurements revealed that the ECSA-corrected CO₂ adsorption amount increased in the sequence of In foil < surface-oxidized In foil < S0–In (Supplementary Fig. 17), and this agrees with the sequence of FE of formate. Therefore, we propose that the co-existence of oxide species as well as the nanostructure morphology may account for the high FE of formate during the CO₂RR over the S0–In catalyst probably by enhancing the adsorption of CO₂ onto catalyst surfaces” (*please see from Page 10 to Page 11-Paragraph 1*).

Specific Comments 1 and 2: The comparison in Fig. 1b is unfair, since the ECSA is not considered. I recommend Suppl. Fig. 2 to be transferred to the main text with the ECSA correction implemented. Please keep the use of bars and dots consistent in the supplementary data and manuscript.

Reply and actions taken: We agree with the reviewer that the comparison based on ECSA should be provided. Thus, we have changed the related Figures in the revised manuscript. The ECSA-corrected formation rates of products over In foil and the S2–In catalyst at different applied potentials have been displayed in Fig. 1c in the revised manuscript to replace previous Fig. 1b for a better comparison. We have moved the previous Fig. 1b, which compared the formation rate of products without ECSA correction over In foil and the S0–In catalyst to Supplementary Information (Supplementary Fig. 3c).

As shown in Fig. 1c in the revised manuscript, even after the ECSA correction has been implemented, the S2–In catalyst still possesses higher formation rate of formate than In foil. It is noteworthy that the superiority of the S2–In catalyst for the formation of all products (HCOO⁻, CO and H₂) became less significant at a potential more negative than -1.03 V versus RHE probably due to the mass-transport limitation at a high current density. To describe these results, we have revised the following sentences in the revised manuscript: “*The performance of the S₂–In catalyst was further compared with that of In foil, a reference catalyst, at different potentials. The S₂–In catalyst exhibited significantly higher current density, FE and formation rate of formate than In foil at each potential (Supplementary Fig. 3). For a better comparison, we have normalized the formation rate of formate based on the electrochemical surface area (ECSA) (Supplementary Table 2), which was determined by double-layer capacitance (C_{dl}) method (Supplementary Fig. 5). The S₂–In catalyst exhibited higher normalized formation rate of formate than In foil (Fig. 1c). The superiority of the S₂–In catalyst for the formation of all products (HCOO⁻, CO and H₂) became less significant at potentials more negative than -1.03 V versus RHE probably because of the mass-transport limitation at a high current density*” (*please*

see Page 6, Lines 12-21).

Specific Comment 3: A parallel study on the modification of an In foil with sulfur would help disentangle the nanostructuring effect and the porosity of the electrode from the effect sulfur brings to In surfaces.

Reply and actions taken: We thank the reviewer for this constructive comment. In fact, in our manuscript, to disentangle the effect of various factors, we have added sulfur onto the S0–In catalyst, which is composed of In nanoparticles without sulfur, by impregnation in Na₂S with different concentrations, and the obtained S-impregnated S0–In series of catalysts with different sulfur contents have been studied for the CO₂RR. The result has confirmed the role of sulfur in enhancing the formation rate of formate (Supplementary Fig. 18a).

As requested by the reviewer, we have further carried out similar studies using In foil as the base to further exclude the influences of nanostructures and surface oxides species. In the experiment, sulfur was added by impregnation of In foil in Na₂S aqueous solutions with different concentrations, and the obtained S-impregnated In foil catalysts have different sulfur contents ranging from 0-7.0 mol% (measured by Auger electron spectroscopy). The catalytic results of the S-impregnated In foil catalysts have been displayed in Supplementary Fig. 18b in the revised manuscript. Similar to the S0–In, the presence of sulfur with a proper content on the surface of In foil could also accelerate the formation of formate. The formation rate of formate in CO₂RR increased upon increasing the content of sulfur up to 2.2 mol%, and a further increase in sulfur content decreased the formation rate of formate. The FE of formate did not change significantly at the same time. These observations are similar to those observed for the S-impregnated S0–In series of catalysts. We have completely rewritten a paragraph to describe these results and please see the following paragraph in the revised manuscript: *“To demonstrate the intrinsic role of sulfur, we have modified the S0–In catalyst with sulfur by a simple impregnation method. The obtained S-impregnated S0–In catalysts with sulfur contents ranging from 0 to 7.1 mol% have been used for CO₂RR. The formation rate of formate increased with an increase in sulfur content up to 2.6 mol% and then decreased (Supplementary Fig. 18a). We performed similar studies using In foil to further exclude the influences of nanostructures and surface oxide species. The electrocatalytic CO₂RR using S-impregnated In foil catalysts with sulfur contents of 0-7.0 mol% showed similar dependences of catalytic behaviors on sulfur content (Supplementary Fig. 18b). The presence of sulfur on In foil with a proper content (≤ 2.2 mol%) significantly enhanced the formation rate of formate, although the value of formation rate was much lower as compared with that on the S-impregnated S0–In series of catalysts. The change in FE*

of formate with sulfur content was less significant for both series of catalysts (Supplementary Fig. 18). These results are consistent with those observed for the S–In series of catalysts (Fig. 1a) and confirm that the sulfur species on In surfaces contributes to promoting the activity of CO₂RR to formate” (please see Page 11, Paragraph 2).

Specific Comment 4 In spite of the consistent evidence of In³⁺ and In⁰ in the catalyst even under reaction conditions, authors do not comment further. There is reported evidence of the role of In hydroxide species in the electrocatalytic reduction of CO₂ (ref. 23, used in the text to assign XPS peaks, or *ACS Catal.* 2016, **6**, 6265–6274.)

Reply and actions taken: We thank the reviewer for this pertinent comment. We have added O1s XPS spectra for the S–In series of catalysts in Supplementary Fig. 13 and the O1s XPS spectrum for the S2–In catalyst after reaction in Supplementary Fig. 14c in the revised manuscript. It is known that the binding energies of O1s in In₂O₃ and In(OH)₃ are ~530.6 and ~531.9 eV, respectively (*J. Appl. Phys.* **51**, 2620-2624 (1980); *Langmuir* **30**, 7593-7600 (2014)). In our case, the O1s binding energy values for the S–In catalysts, either before or after reaction, are located at 530.6 eV (Supplementary Fig. 13 and 14c). Furthermore, it has been pointed out that In(OH)₃ on In metal electrode could exist in an electrolyte with a pH of 4.4, but In(OH)₃ would be transformed to In₂O₃ by changing the pH of electrolyte from 4.4 to 7.7 (*Langmuir* **30**, 7593-7600 (2014)). Generally, In(OH)₃ would be transformed to In₂O₃ at a pH of 6.8 (*Encyclopedia of Electrochemistry of the Elements*, 1976, **Volume VI**, 1-31). We carried out the electrocatalytic experiments in a buffer of CO₂-saturated 0.5 M KHCO₃ with a pH of 7.2, favoring In₂O₃ but not In(OH)₃ species. These facts allow us to consider that the In³⁺ species on our S–In catalysts are mainly in In₂O₃ state.

As already mentioned in the **Reply to General Comment 3**, the S0–In catalyst with In₂O₃ on its surfaces shows higher FE of formate during CO₂RR than In foil, although the ECSA-corrected activities (total formation rates of all products) are similar. To understand whether In₂O₃ present on In surfaces is really beneficial to the FE of formate, we have pretreated In foil in air at 250 °C to generate a coverage of In₂O₃ on metallic In foil surfaces. The electrocatalytic CO₂RR result showed that the FE of formate increased on the surface-oxidized In foil (Supplementary Fig. 16). Our CO₂-adsorption measurements revealed that the ECSA-corrected CO₂ adsorption amount increased in the sequence of In foil < surface-oxidized In foil < S0–In (Supplementary Fig. 17), and this agrees with the sequence of the FE of formate. Therefore, we propose that the co-existence of oxide species contributes to the high FE of formate during the CO₂RR by accelerating CO₂ adsorption. The following

sentences have been added in the revised manuscript to describe these results and discussion: “Moreover, our XPS measurements reveal that a small fraction of In^{3+} (i.e., In_2O_3) species co-exists with metallic In on the S0–In surface. The nanostructured morphology and the presence of oxidized species on metal catalysts were reported to be beneficial to $\text{CO}_2\text{RR}^{23,28-30}$. In particular, $\text{In}(\text{OH})_3$ was proposed to play a crucial role in the formation of formate or $\text{CO}^{23,30}$. Our XPS results indicated the co-existence of In_2O_3 but not $\text{In}(\text{OH})_3$ with In^0 in our case. To understand the role of surface oxidized species, we further pretreated In foil in air at 250 °C for 3 h to generate a coverage of In_2O_3 on In surfaces. The electrocatalytic CO_2RR result showed that the FE of formate increased on the surface-oxidized In foil, although the formation rate of all products based on ECSA did not change significantly (Supplementary Fig. 16). Our CO_2 -adsorption measurements revealed that the ECSA-corrected CO_2 adsorption amount increased in the sequence of In foil < surface-oxidized In foil < S0–In (Supplementary Fig. 17), and this agrees with the sequence of FE of formate. Therefore, we propose that the co-existence of oxide species as well as the nanostructure morphology may account for the high FE of formate during the CO_2RR over the S0–In catalyst probably by enhancing the adsorption of CO_2 onto catalyst surfaces” (please see from Page 10-Line 11 to Page 11-Line 3).

Specific Comment 5: Though reasonable, the mechanistic reason aiming to explain the role of sulfur is loosely bound to the DFT calculations. Authors observe a large predominance of the (101) facet in XRD and also in TEM analysis, in contrast to the (100) selected for the theoretical study. Authors must justify this selection. In parallel, authors claim water acting as the proton donor as the key step, but DFT calculations leave this step aside as they assume H^+ in solution. Though aware of the limitations of DFT, authors should find a more solid background for their hypothesis to strengthen their conclusions.

Reply and actions taken: We thank the reviewer for this constructive comment. We have revised the DFT calculation model, by using (101) facet as the model and H_2O as the proton donor. The recalculated results have been displayed in Fig. 3, Supplementary Fig. 21 and Supplementary Table 4 to replace the previous results using (100) facet and H^+ for calculation. Although the absolute values are different from the previous results in the old manuscript, the essence and the changing trend remain the same.

We have also modified the following sentences in the main text in the revised manuscript to describe the newly calculated DFT results: “For the HCOOH pathway, the Gibbs free energies (ΔG) for formations of HCOO^* and HCOOH^* are 0.29 and 0.67 eV on indium only surfaces (Fig. 3b). The presence of sulfur on indium

significantly decreases the corresponding Gibbs free energies for HCOO and HCOOH* to -0.16 and 0.10 eV, respectively (Fig. 3b)” (please see Page 14, Lines 1-4); “For the CO pathway, the Gibbs free energies for the formation of *COOH are 1.49 and 0.82 eV on pure and sulfur-doped indium surfaces, respectively (Fig. 3c)” (please see from Page 14, Lines 6-8); “The formation energy of H* species is 0.21 eV on sulfur sites of sulfur-doped indium, significantly lower than that on indium sites of sulfur-doped indium (0.69 eV) and pure indium (0.82 eV) (Fig. 3d)” (please see from Page 14, Paragraph 2, Lines 2-4).*

Specific Comment 6: The shown enhancement in the formate production rate upon modification with Pt is within the error bars and, as authors acknowledge, was not a successful attempt. I suggest to move it to the supplementary information.

Reply and actions taken: As suggested by the reviewer, we have moved the figure of modification with Pt from Fig. 4a in the previous version of manuscript into Supplementary Fig. 22 in the revised manuscript.

Response to Reviewer 3

General Comments: This is a rather interesting paper that reports for the first time that the presence of sulfide on the surface of an indium (or other heavy p-group metal) electrode enhances the electrodes electrocatalytic activity toward CO₂ reduction to formate in a basic aqueous electrolyte. The authors report a high faradaic efficiency that is maintained even at high current densities – a finding the authors claim is unique. In fact, though rare, other indium-based systems will do this also, and are of industrial interest. (See for example: Journal of CO₂ Utilization 7, (2014) p1–5) Nonetheless, the authors finding that sulfur treated electrodes behavior in this manner is intriguing and novel from the chemical point of view. This finding leads the authors to suggest that the role of the surface sulfide is to activate the formation of a surface hydrogen atom formed from a water ligated to a supporting electrolyte cation. To probe this possibility, the researchers cleverly exam the effect of varying the alkali cation present in the electrolyte on the formation of formate. They report that as one drops down the first column of the periodic table (reducing the number of waters ligated to the cation) that the catalytic efficiency improves, and argue that a smaller hydrated cation will interact more strongly with the surface sulfide anion. The papers conclusions are supported by an ample set of experimental data covering issues of surface science, electrochemistry and quantum chemistry simulation. Yet, given the extensive experiments and thought that has gone into this paper there are some surprising omissions.

Reply and actions taken: We appreciate the positive evaluation by this reviewer.

We have read the reference (J. CO₂ Util. 7, 1-5 (2014)) provided by the reviewer with interest. We found that the difference between this reference and our work is clear, although both studies have used indium-based electrocatalysts. In the reference, Bocarsly and co-workers reported a solar cell-powered electrochemical CO₂RR using In as a cathode material. The work used a stack of three electrochemical flow cells in series, with which the current densities of the three cells could add up. The electrochemical flow cell the reference employed could also increase the current density by lowering the mass transportation. Because of the advantages of cell design, Bocarsly and co-workers achieved a 67% FE of formate at current density of ~100 mA cm⁻². On the other hand, we have obtained 85% FE of formate at current density of ~100 mA cm⁻² by using a conventional aqueous H-type electrochemical cell and well-designed electrocatalyst. In short, the reference work reported system design and demonstration, whereas our present work focused on fundamental studies to develop efficient electrocatalysts. We believe that the design of electrochemical cells like that in the reference will further enhance the performance of our catalyst in the future. We have added this reference as Ref. 25 in the revised manuscript. The following sentence in the Introduction section has been modified in the revised manuscript: “*However, the activity of indium catalysts is usually low (current density < 6 mA cm⁻²),^{10,23,24} although the design of special electrochemical cell can enhance the activity²⁵” (Please see Page 4, Paragraph 2, Lines 2-4).*

Our responses to the detailed issues raised by the reviewer and the corresponding revisions are described as follows.

Comment 1: A ¹³CO₂ experiment is not reported to demonstrate that CO₂ is in fact the source of the observed products. This is considered a standard control in the area of CO₂ electrochemistry at this point.

Reply and actions taken: We thank the reviewer for this pertinent comment. We have carried out the ¹³CO₂ labelling experiment for CO₂RR using the S2-In catalyst at -0.98 V versus RHE. The products were analyzed by ¹H and ¹³C NMR. The obtained results have been displayed in Supplementary Fig. 2 in the revised manuscript. In the ¹H NMR spectrum, a doublet was observed at 8.5 ppm, which was attributable to the proton coupled to ¹³C of H¹³COO⁻. A signal at 168.5 ppm was observed in the ¹³C NMR spectrum, which could be ascribed to H¹³COO⁻ (*Nature* **529**, 68-71 (2016)). These results confirm that formate is formed from CO₂ but not from other carbon sources.

We have added the following new paragraph to describe these results in the revised manuscript: “*We conducted ¹³CO₂ labelling experiments for the S2-In*

catalyst. The products obtained at a potential of -0.98 V versus RHE were analysed by ^1H and ^{13}C nuclear magnetic resonance (NMR) spectroscopy. A ^1H NMR doublet was observed at 8.5 ppm, which was attributable to the proton coupled to ^{13}C in $\text{H}^{13}\text{COO}^-$ (Supplementary Fig. 2a). A signal at 168.5 ppm was observed in the ^{13}C NMR spectrum, which could be ascribed to $\text{H}^{13}\text{COO}^-$ (Supplementary Fig. 2b)¹¹. These observations confirm that formate is formed from CO_2 reduction” (please see Page 5, Paragraph 2).

Comment 2: A pH dependence is not undertaken, even though the author’s mechanism requires a basic electrolyte and a pH study would certainly shed more light on the mechanism of CO_2 reduction.

Reply and actions taken: We agree with the reviewer that the study on pH-dependence may provide further insights into the reaction mechanism. Thus, we have investigated the effect of pH on electrocatalytic CO_2RR using the S0–In and S2–In catalysts.

We used three different electrolytes, i.e., K_2HPO_4 , KHCO_3 and K_2SO_4 , to regulate the pH value, because it is known that the local pH at the cathode/electrolyte interface increases in the following sequence: $\text{K}_2\text{HPO}_4 < \text{KHCO}_3 < \text{K}_2\text{SO}_4$ (*Angew. Chem. Int. Ed.* 2016, **55**, 6680-6684. *J. Chem. Soc. Faraday Trans.* 1989, **85**, 2309-2326). As reported in *Angew. Chem. Int. Ed.* and *J. Chem. Soc. Faraday Trans.*, this difference arises from the difference in the buffer capacity of the three electrolytes. It is widely accepted that the release of OH^- from CO_2RR ($\text{CO}_2 + \text{H}_2\text{O} + 2\text{e}^- \rightarrow \text{HCOO}^- + \text{OH}^-$; $\text{CO}_2 + \text{H}_2\text{O} + 2\text{e}^- \rightarrow \text{CO} + 2\text{OH}^-$; $\text{H}_2\text{O} + \text{e}^- \rightarrow 0.5\text{H}_2 + \text{OH}^-$) can cause a non-equilibrium local high-pH region near the cathode. CO_2 -saturated 0.5 M K_2HPO_4 (pH = 6.8) and CO_2 -saturated 0.5 M KHCO_3 (pH = 7.2) can easily neutralize OH^- , maintaining the local pH in the sequence of $\text{K}_2\text{HPO}_4 < \text{KHCO}_3$, whereas the non-buffer property of K_2SO_4 leads to high local pH values (*Angew. Chem. Int. Ed.* 2016, **55**, 6680-6684. *J. Chem. Soc., Faraday Trans.* 1989, **85**, 2309-2326). Our electrocatalytic CO_2RR results showed that the formation rate and EF of formate increased in the sequence of $\text{K}_2\text{HPO}_4 < \text{KHCO}_3 < \text{K}_2\text{SO}_4$ over the S0–In and S2–In catalysts (Supplementary Fig. 20), indicating that a higher local pH environment favors the formation of formate. As compared to the S0–In catalyst, the S2–In exhibited higher formation rate and EF of formate using all the three electrolytes. Furthermore, we calculated the ratio of formate rates of formate for the S2–In and S0–In catalysts, which was denoted as $\text{Rate}_{\text{S2-In}}/\text{Rate}_{\text{S0-In}}$. The $\text{Rate}_{\text{S2-In}}/\text{Rate}_{\text{S0-In}}$ value increases from 1.4 to 1.9 and further to 2.1 upon changing the electrolyte from K_2HPO_4 to KHCO_3 and further to K_2SO_4 (Supplementary Fig. 20). This suggests that the role of sulfur in enhancing the formation of formate is

more significant at a higher pH value. This confirms that the modification by sulfur enhances the formation of formate by accelerating the activation of H₂O, which becomes more difficult at a higher pH.

We have added the following new paragraph in the revised manuscript to describe the results and discussion mentioned above: *“To obtain further evidence for the role of H₂O activation in CO₂RR, we have investigated the effect of pH of electrolyte on electrocatalytic CO₂RR over S0–In and S2–In catalysts. Three different electrolytes, i.e., K₂HPO₄, KHCO₃ and K₂SO₄, were employed to regulate the pH value, because it is known that the local pH at the cathode/electrolyte interface increases in the following sequence: K₂HPO₄ < KHCO₃ < K₂SO₄^{34,35}. Our electrocatalytic results show that the formation rate and FE of formate increase in the sequence of K₂HPO₄ < KHCO₃ < K₂SO₄ over both catalysts (Supplementary Fig. 20), indicating that a higher local pH environment favors the formation of formate. As compared to the S0–In, the S2–In catalyst exhibited higher formation rate and FE of formate using all the three electrolytes. Furthermore, the ratio of formation rates of formate for the S2–In and S0–In catalysts, i.e., Rate_{S2–In}/Rate_{S0–In}, increased from 1.4 to 1.9 and further to 2.1 upon changing the electrolyte from K₂HPO₄ to KHCO₃ and further to K₂SO₄ (Supplementary Fig. 20a). This suggests that the role of sulfur in enhancing the formation of formate is more significant at a higher pH value. This supports our speculation that the sulfur modification enhances formate formation by accelerating the activation of H₂O, which becomes more difficult at a higher pH^{32,33}”* (please see from Page 12-the last paragraph to Page 13-Paragraph 1).

Comment 3: MOST IMPORTANTLY: Though the authors cite at least one paper (reference 20) that argues that surface oxides are responsible for the electrocatalytic activity of In with regard to CO₂, the authors never mention the existence of a surface oxide or what its role might be in their system. Given, both the electrochemistry and surface spectroscopy carried out by the authors, it would be impossible to miss the existence of a surface oxide. Thus, this is either a serious omission, i.e. they elected to not report the presence of surface oxides, or a chemical miracle – adding a submonolayer of sulfide to an indium surface protects it from oxide formation. Given the intense reactivity of “normal” indium with air to form an instantaneous oxide coating – I doubt that the sulfur is suppressing this reaction. If it is, then they have a much bigger finding in the area of corrosion chemistry than the CO₂ chemistry they are reporting. Their lack of reporting the nature of the surface oxide therefore throws the whole study into doubt. It also challenges their DFT study, since their model does not consider the presence or role of a surface oxide. Once this point is dealt with honestly, I will enthusiastically support the publication of this work.

Reply and actions taken: We appreciate very much this constructive comment from the reviewer. In our previous version of manuscript, we focused on the role of sulfur and did not pay attention to the surface oxide species on In surfaces. We totally agree with the reviewer that the discussion on the possible influence of oxide species on the CO₂RR over S–In catalysts is very important.

First, we have checked the presence of oxide species on In surface by EDS and XPS. As pointed out by the reviewer, oxide species really exist on our S–In catalyst surfaces. The EDS elemental mapping for oxygen and the line-scan EDS containing oxygen for the In–S₂ catalyst have been added in Fig. 2b and Supplementary Fig. 11 in the revised manuscript. The O1s XPS spectra for the S–In series of catalysts have been displayed in Supplementary Fig. 13. From these results, it becomes clear that, together with sulfur, oxygen is also homogeneously distributed over In particles in the catalysts. The O1s XPS result showed that the binding energy of O1s was 530.6 eV, which could be attributed to O²⁻ of In₂O₃ (*J. Appl. Phys.* **51**, 2620-2624 (1980), *Langmuir* **30**, 7593-7600 (2014)). In short, our characterization results indicate that In₂O₃ species co-exist with metallic In on the S–In series of catalysts. Based on these results, we have modified the following sentence in the revised manuscript: “*The energy-dispersive X-ray spectroscopy (EDS) analysis for the S₂–In catalyst indicated that In, S and O elements existed in the catalyst, and these elements were distributed uniformly over the catalyst particle (Fig. 2b and Supplementary Fig. 11)*” (please see Page 8, Paragraph 2, Lines 1-3). We have also added the following sentences to describe the O1s XPS result in the revised manuscript: “*The O 1s spectra displayed a peak at 530.6 eV, which could be assigned to O²⁻ in In₂O₃ (Supplementary Fig. 13)^{23,27}. Thus, In₂O₃ species co-exist with metallic In on the catalyst surface in addition to sulfide species*” (please see Page 9, Lines 3-5).

Second, we found that the S₀–In catalyst (without sulfur but possessing oxide species on its surfaces) showed higher FE of formate than metallic In foil, although the activities (formation of all products) based on electrochemical surface area (ECSA) were almost the same for the two catalysts. Thus, the co-existing In₂O₃ may be beneficial to FE of formate in CO₂RR. To further understand whether In₂O₃ present on In surfaces plays a role in CO₂RR, we have pretreated In foil in air at 250 °C to generate a coverage of In₂O₃ on the surface of metallic In foil. The electrocatalytic CO₂RR result showed that the FE of formate increased on the surface-oxidized In foil (Supplementary Fig. 16). Our CO₂-adsorption measurements revealed that the ECSA-corrected CO₂ adsorption amount increased in the sequence of In foil < surface-oxidized In foil < S₀–In (Supplementary Fig. 17), and this agrees with the sequence of FE of formate. Therefore, we propose that the co-existence of oxide species contributes to the high FE of formate during the CO₂RRs. The following new

paragraph has been added to describe these results and discussion in the revised manuscript: “*Our present work has demonstrated that the sulfur-modified In catalyst is very promising for electrocatalytic reduction of CO₂ to formate. To understand the role of sulfur more deeply, it is necessary to disentangle different factors that may contribute to CO₂RR in the present system. Our results show that the S0–In catalyst without sulfur fabricated by electroreduction of In₂O₃ precursor growing on carbon fibers exhibits higher FE of formate than In foil (Fig. 1a and Supplementary Fig. 16a). The activity of the S0–In catalyst, expressed by the ECSA-corrected formation rate of all products (including HCOO[−], CO and H₂), is almost the same with that of In foil (Supplementary Fig. 16b). The S0–In catalyst exhibits nanoparticulate morphology with an average diameter of 128 nm, whereas In foil has smooth surfaces (Supplementary Fig. 8). Moreover, our XPS measurements reveal that a small fraction of In³⁺ (i.e., In₂O₃) species co-exists with metallic In on the S0–In surface. The nanostructured morphology and the presence of oxidized species on metal catalysts were reported to be beneficial to CO₂RR^{23,28-30}. In particular, In(OH)₃ was proposed to play a crucial role in the formation of formate or CO^{23,30}. Our XPS results indicated the co-existence of In₂O₃ but not In(OH)₃ with In⁰ in our case. To understand the role of surface oxidized species, we further pretreated In foil in air at 250 °C for 3 h to generate a coverage of In₂O₃ on In surfaces. The electrocatalytic CO₂RR result showed that the FE of formate increased on the surface-oxidized In foil, although the formation rate of all products based on ECSA did not change significantly (Supplementary Fig. 16). Our CO₂-adsorption measurements revealed that the ECSA-corrected CO₂ adsorption amount increased in the sequence of In foil < surface-oxidized In foil < S0–In (Supplementary Fig. 17), and this agrees with the sequence of FE of formate. Therefore, we propose that the co-existence of oxide species as well as the nanostructure morphology may account for the high FE of formate during the CO₂RR over the S0–In catalyst probably by enhancing the adsorption of CO₂ onto catalyst surfaces” (please see Page 9, Paragraph 2).*

Third, to exclude the influence of the co-existing In₂O₃ species and to gain information on the intrinsic role of sulfur, we have further added sulfur onto the S0–In catalyst, which is composed of In nanoparticles without sulfur, and In foil by impregnation in Na₂S with different concentrations. The obtained S-impregnated S0–In series and S-impregnated In foil series of catalysts with different sulfur contents have been studied for the CO₂RR. The result has confirmed the role of sulfur in enhancing the formation rate of formate (Supplementary Fig. 15). The following new paragraph has been added to describe these results and discussion in the revised manuscript: “*To gain more insights into the role of sulfur, we have modified the S0–In catalyst with sulfur by a simple impregnation method. The obtained S-impregnated*

S0–In catalysts with sulfur contents ranging from 0 to 7.1 mol% have been used for CO₂RR. The formation rate of formate increased with an increase in sulfur content up to 2.6 mol% and then decreased with a further increase in sulfur content (Supplementary Fig. 17a). We performed similar studies using In foil to further exclude the influences of nanostructures and surface oxide species. The electrocatalytic CO₂RR using S-impregnated In foil catalysts with sulfur contents of 0-7.0% showed similar dependences of catalytic behaviors on sulfur content (Supplementary Fig. 17b). The presence of sulfur on In foil with a proper content (≤ 2.2 atom%) significantly enhanced the formation rate of formate, although the value of formation rate was much lower as compared with that on the S-impregnated S0–In series of catalysts. The change in FE of formate with sulfur content is less significant for both series of catalysts. These results are consistent with those observed for the S–In series of catalysts by changing the sulfur content (Fig. 1a) and confirm that the sulfur species on In surfaces contributes to promoting the activity of CO₂RR to formate” (please see from Page 10 to Page 11-Line 3).

Finally, for the DFT calculation, to be honest, it is a tough and long-term project to establish a suitable model that takes metal, metal oxide and sulfide into consideration simultaneously. This is beyond our calculation ability at the current stage. Since we have already disentangled different factors in our revised manuscript, we think that any conclusions cannot be spoiled without such a DFT study.

Comment 4: Finally, I note that the experiments involving the addition of sodium sulfide to the electrolyte are not sufficiently clear to reproduce. Is the sulfide present in the CO₂ purged electrolyte or does the sulfide exposure involve a pretreatment of the electrode?

Reply and actions taken: We thank the reviewer for this pertinent comment. The purpose of these experiments is to exclude the influence of the co-existing oxide species or nanostructures in the S–In series of catalysts and to gain insights into the intrinsic role of sulfur. Thus, we have attempted to add sulfur to the already prepared catalyst but not add sulfur in catalyst-synthesis stage (like that used for the S–In series of catalysts). As compared to adding Na₂S into the electrolyte, we now think that the addition of sulfur species directly onto the S0–In or In foil is more convincing. Therefore, we have adopted a simple impregnation method to add Na₂S onto the S0–In without sulfur and In foil. In the experiment, sulfur was simply added by impregnation of S0–In or In foil in Na₂S aqueous solutions with different concentrations, and the obtained S-impregnated S0–In and In foil catalysts have different sulfur contents ranging from 0-7.0 mol% (measured by Auger electron spectroscopy). The catalytic results of the S-impregnated S0–In and In foil have been

displayed in Supplementary Fig. 18 in the revised manuscript. The presence of sulfur with a proper content on the surface of S0–In or In foil could accelerate the formation of formate. These results provide further evidence that the presence of sulfur enhances the activity of CO₂RR to formate.

To make this point clear in the revised manuscript, we have added the following paragraph in the Methods: “***Fabrication of S-impregnated S0–In and In foil electrocatalysts.*** The S0–In catalyst, which was fabricated above and did not contain sulfur, was also modified with sulfur by an impregnation method. The S0–In catalyst was impregnated in Na₂S aqueous solutions with different concentrations (0.10, 0.25, 0.50 and 1.0 mM) for 5 min. Then, the catalyst was dried and was used for the CO₂RR. The obtained samples were denoted as S-impregnated S0–In and the sulfur contents measured by Auger electron spectroscopy were 1.4, 2.6, 5.2 and 7.1 mol%. The S-impregnated In foil samples were prepared by the same procedure. In foil was first etched in 5.0 M HCl for 5 min to remove native oxides or impurities under the protection of N₂ atmosphere. The pretreated In foil was then impregnated in Na₂S aqueous solutions with concentrations of 0.10, 0.25, 0.50 and 1.0 mM for 3 min under the protection of N₂, obtaining S-impregnated In foil samples with sulfur contents of 0.8, 2.2, 4.0 and 7.0 mol%, respectively” (please see Page 19, Paragraph 2, Section Fabrication of S-impregnated S0–In and In foil electrocatalysts). Furthermore, we have completely rewritten a paragraph to describe the results in the main text and please see the following paragraph in the revised manuscript: “To demonstrate the intrinsic role of sulfur, we have modified the S0–In catalyst with sulfur by a simple impregnation method. The obtained S-impregnated S0–In catalysts with sulfur contents ranging from 0 to 7.1 mol% have been used for CO₂RR. The formation rate of formate increased with an increase in sulfur content up to 2.6 mol% and then decreased (Supplementary Fig. 18a). We performed similar studies using In foil to further exclude the influences of nanostructures and surface oxide species. The electrocatalytic CO₂RR using S-impregnated In foil catalysts with sulfur contents of 0-7.0 mol% showed similar dependences of catalytic behaviors on sulfur content (Supplementary Fig. 18b). The presence of sulfur on In foil with a proper content (≤ 2.2 mol%) significantly enhanced the formation rate of formate, although the value of formation rate was much lower as compared with that on the S-impregnated S0–In series of catalysts. The change in FE of formate with sulfur content was less significant for both series of catalysts (Supplementary Fig. 18). These results are consistent with those observed for the S–In series of catalysts (Fig. 1a) and confirm that the sulfur species on In surfaces contributes to promoting the activity of CO₂RR to formate” (please see Page 11, Paragraph 2).

Reviewers' comments:

Reviewer #1 (Remarks to the Author):

Wang and co-workers report the discovery of a promoting effect of chalcogen doping on electrocatalytic CO₂ reduction to formate. The authors have obviously done a very thorough study. However, there is strong deficiency in data interpretation. Therefore, extensive revision is needed in order to make this manuscript suitable for publication. Detailed comments are given below:

(1) The mass transport limiting current estimated for CO₂ reduction in a H-cell configuration under ambient condition is shockingly high. To obtain this value, the authors assume a laminar flow under rotating disk conditions for approximation and use a rotation rate as high as >2000 rpm, a rate that is hardly achievable with a H-cell configuration using a planar electrode (either porous or nonporous). The mass transport limiting current obtained by others is at least a factor of 2 smaller. Furthermore, the authors should not compare their value with the one obtained with a nano-needle where mass transport is governed by radial diffusion.

(2) The authors stated the role of water by referring to the reaction described in eq 1. This is completely misleading since eq 1 is an overall reaction. A reaction scheme with all detailed elementary reaction steps in particular the rate determining step are needed for the readers to understand the significant role of water. Furthermore, it is confusing to me why water should be the source for protons during CO₂ reduction. Shouldn't it be HCO₃⁻, the strongest acid in this medium?

(3) It is unclear to me how the results in Fig S17 were obtained. Were they obtained under gas phase conditions? If this is the case, how can they reflect the situation in an electrolyte medium?

(4) It is unclear to me based on the DFT calculation why S doping promoted CO₂ RR on In but not HER.

(5) In supplementary Fig. 10, why is the S₀-In has a larger interplanar spacing than the other materials?

Reviewer #2 (Remarks to the Author):

The authors made a considerable effort to modulate the main messages and to provide additional experimental evidence supporting their main claims.

To my understanding, the main claim of sulfur acting as a promoter of the activity by a mechanism where water may act as the proton source is now solid enough to be considered seriously by the research community, and thus valid to inspire new work in multicomponent electrocatalysts for this and other reactions. However, at this point, authors should discuss the very recently published experimental-theoretical study on the mechanism behind the promotion of the formate route by sulfur species on Cu by Pérez-Ramírez et al. (J. Phys. Chem. Lett. 2018, doi:10.1021/acs.jpcclett.8b03212), which suggests a very different mechanism to the one proposed by the authors.

Additional experiments performed by the authors are consistent with the overall picture, which points out to the strong tendency of sulfur to promote the formate route almost independently of the structure and chemical state of the base In. Authors also made the effort to shed light on the importance of oxidic In species on the reaction, though failed to consider them in the DFT study. In

general, even though the picture is still fragmented, their findings are robust and have the potential to act as strong seeds for further developments in terms of novel catalysts and understanding.

I, therefore, recommend publication upon consideration of previous comments.

Responses to Reviewers

Response to Reviewer 1

General comments: Wang and co-workers report the discovery of a promoting effect of chalcogen doping on electrocatalytic CO₂ reduction to formate. The authors have obviously done a very thorough study. However, there is strong deficiency in data interpretation. Therefore, extensive revision is needed in order to make this manuscript suitable for publication. Detailed comments are given below:

Reply and actions taken: We appreciate the critical comments raised by this reviewer. We have performed additional experiments to answer the questions and comments raised by this reviewer. We have also made major revisions based on the questions and comments by the reviewers and our new experimental results. Our replies to the comments and the corresponding revisions are described as follows.

Comment 1: The mass transport limiting current estimated for CO₂ reduction in a H-cell confusion under ambient condition is shockingly high. To obtain this value, the authors assume a laminar flow under rotating disk conditions for approximation and use a rotation rate as high as >2000 rpm, a rate that is hardly achievable with a H-cell configuration using a planar electrode (either porous or nonporous). The mass transport limiting current obtained by others is at least a factor of 2 smaller. Furthermore, the authors should not compare their value with the one obtained with a nano-needle where mass transport is governed by radial diffusion.

Reply and actions taken: In the previous version of revision, we explained that the maximum current density for CO₂RR in our study (86 mA cm⁻²) approached the current density under the mass-transport limitation. The current density under mass-transport limitation was evaluated using the equation of $j_{\text{limit}} = (n \times F \times D \times C) / \delta$. We calculated δ , i.e., the diffusion layer thickness, using the equation of $\delta = (1.61 \times D^{1/3} \times \nu^{1/6}) / \omega^{1/2}$, which was derived from a rotating disk electrode model. The rotation rate used for the calculation of ω is a key parameter determining the diffusion layer thickness. We adopted 2000 rpm for calculation, which was the stirring speed of magnetic stirrer achieved with a magnetic agitator (IKA-Big squid) used in our cathodic compartment of H-cell.

It is noteworthy that the rotation rate of 2000 rpm is widely used for the rotating disk electrode. From the textbook (please see for example: *A First Course in Electrode Processes*, Page 163-166, Figure 7.4-7.6, RSC publishing, Derek Pletcher 2009), we know that the liner relationship of diffusion layer can be obtained by voltammograms at rotation rates ranging from 400-3600 rpm for the rotating disk

electrode. Many recent papers reported electrocatalytic reactions using rotation rates of >2000 rpm (for examples, 3600 rpm in *Angew. Chem. Int. Ed.* **48**, 4386-4389 (2009); 3025 rpm in *ACS Catal.* **6**, 4720-4728 (2016)).

To gain the information of diffusion layer thickness under our reaction conditions, we have compared currents between the agitation with magnetic stirrer and the rotation of rotating disk electrode in our H-cell by using linear sweep voltammetry (LSV) measurement. We used a rotating glassy carbon disc electrode doped with Pt of $3.75 \mu\text{g cm}^{-2}$ as a working electrode at the same position of cathode for our CO₂RR measurements. The reduction of K₃Fe(CN)₆ was chosen to probe the diffusion layer thickness because of its electrochemical reversibility, meaning that the reduction of K₃Fe(CN)₆ is facile so that the observed rate is limited only by mass transfer regardless of the applied overpotential (*ACS Catal.* **8**, 6560-6570 (2018)). We performed the LSV measurement in 10 mM K₃Fe(CN)₆ solution with 0.5 M KHCO₃ as an electrolyte at a scan rate of 10 mV s⁻¹ from 1.4 to -0.2 V vs. RHE. The current-potential curve was first recorded at the stirring speed of 2000 rpm by magnetic stirrer. Then, the current-potential curves of rotating disk electrode at different rotation rates ranging from 500 to 2000 rpm were recorded to fit the current-potential curve obtained by the magnetic stirrer agitation. The comparison reveals that the current-potential curve for the stirring speed of 2000 rpm is quite close to that for the rotating disk electrode with a rotation rate of 1800 rpm (please see Supplementary Fig. 26). This result allows us to conclude that the stirring speed of 2000 rpm in our case is comparable to the rotation rate of 1800 rpm in the rotating disk electrode.

Based on the experiment result described above, we have recalculated the diffusion layer thickness by using the equation of $\delta = (1.61 \times D^{1/3} \times v^{1/6}) / \omega^{1/2}$ with a rotation rate of 1800 rpm. The diffusion layer thickness is calculated to be 14.8 μm . Further, the current density under mass-transport limitation for CO₂RR has been recalculated to be 90 mA cm⁻² based on the equation of $J_{\text{limit}} = (n \times F \times D \times C) / \delta$. Therefore, our conclusion that the maximum current density for CO₂RR observed in our study (86 mA cm⁻²) approaches the current density under the mass-transport limitation remains correct.

Moreover, we found that there are a couple of papers that have reported the current density for CO₂RR under mass-transfer limitation. Bao and co-workers have shown that the J_{limit} is 94 mA cm⁻² by using a rotation rate of 2000 rpm and they have used the same H-cell with the magnetic stirrer agitation with us (*Energy Environ. Sci.* **11**, 1204-1210 (2018)). Zhang and co-workers reported a J_{limit} of 45 mA cm⁻² by adopting a rotation speed of 500 rpm and they have regarded this value as a moderate one (*Angew. Chem. Int. Ed.* **56**, 505-509 (2017)).

Based on the results and discussion described above, we have made the following revisions in the revised manuscript:

- (1) We have corrected the maximum value evaluated by assuming the mass-transportation from 94 to 90 mA cm⁻² (*please see Page 6, Line 10*).
- (2) We have added the following sentence in the Method section (Electrochemical measurements) in the main text to describe the stirring speed: “*During electrocatalytic reactions, the solution in cathode compartment was vigorously stirred at a speed of 2000 rpm using a magnetic stirrer*” (*please see Page 22, Lines 4-3 from bottom*).
- (3) We have moved the evaluation of current density under mass-transport limitation from Supplementary Information to the Method section (Electrochemical measurements) in the main text (*please see from Page 24-the last paragraph to Page 25-Paragraph 1*). The following sentence has been added as the last sentence of the paragraph to describe the evaluation of the current density under mass-transport limitation: “*To make an accurate evaluation of the diffusion layer thickness, we performed linear sweep voltammetry (LSV) measurements using the rotating disk electrode and the magnetic-stirrer agitation. See Supplementary Method and Supplementary Fig. 26 for details*” (*please see Page 24, Lines 5-3 from bottom*).
- (4) The following new paragraphs have been added in the Supplementary Information (Supplementary Methods): “*Evaluation of the diffusion layer thickness for the current density under mass-transport limitation. As described in Methods, the current density under mass-transport limitation can be evaluated using the following equation¹⁹:*

$$j_{\text{limit}} = (n \times F \times D \times C) / \delta$$

($n = 2$; $F = 96485 \text{ C mol}^{-1}$; $D = 2.02 \times 10^{-9} \text{ m}^2 \text{ s}^{-1}$; $C = 34 \text{ mol m}^{-3}$ at 1 bar and 25 °C)

Here, δ is the diffusion layer thickness for CO₂, which can be estimated from rotating disk electrode model with the following Levich equation¹⁹:

$$\delta = (1.61 \times D^{1/3} \times \nu^{1/6}) / \omega^{1/2}$$

$$(\nu = 1.0 \times 10^{-6} \text{ m}^2 \text{ s}^{-1})$$

Here, ω is the angular frequency of rotation and can be expressed with $2\pi \times$ rotation rate (s^{-1}). Thus, the rotation rate used for the calculation of ω is a key parameter determining the diffusion layer thickness and thus the current density under mass-transport limitation.

To gain the information of diffusion layer thickness under our reaction conditions, we have compared currents between the agitation with magnetic stirrer and the rotation of rotating disk electrode in our H-cell by using linear sweep voltammetry (LSV) measurement (Supplementary Fig. 26a). We used a rotating glassy carbon disc electrode doped with Pt of 3.75 $\mu\text{g cm}^{-2}$ as a working electrode at the

same position of cathode for our CO₂RR measurements. The reduction of K₃Fe(CN)₆ was chosen to probe the diffusion layer thickness because of its electrochemical reversibility, meaning that the reduction of K₃Fe(CN)₆ is facile so that the observed rate is limited only by mass transfer regardless of the applied overpotential²⁰. We performed the LSV measurement in 10 mM K₃Fe(CN)₆ solution with 0.5 M KHCO₃ as an electrolyte at a scan rate of 10 mV s⁻¹ from 1.4 to -0.2 V vs. RHE. The current-potential curve was first recorded at the stirring speed of 2000 rpm by magnetic stirrer. Then, the current-potential curves of rotating disk electrode at different rotation rates ranging from 500 to 2000 rpm were recorded to fit the current-potential curve obtained by the magnetic stirrer agitation. The results have been displayed in Supplementary Fig. 26b. The comparison reveals that the current-potential curve for the stirring speed of 2000 rpm is quite close to that for the rotating disk electrode with a rotation rate of 1800 rpm. This result allows us to conclude that the stirring speed of 2000 rpm in our case is comparable to the rotation rate of 1800 rpm in the rotating disk electrode.

We have calculated the diffusion layer thickness by using the equation of $\delta = (1.61 \times D^{1/3} \times v^{1/6}) / \omega^{1/2}$ with a rotation rate of 1800 rpm. The diffusion layer thickness is calculated to be 14.8 μm . Further, the current density under mass-transport limitation for CO₂RR has been calculated to be 90 mA cm⁻² based on the equation of $J_{\text{limit}} = (n \times F \times D \times C) / \delta$. This value is in agreement with that reported in literature under similar experimental conditions²¹,

(please see Supplementary Information, Pages 23-24).

Comment 2: The authors stated the role of water by referring to the reaction described in eq 1. This is completely misleading since eq 1 is an overall reaction. A reaction scheme with all detailed elementary reaction steps in particular the rate determining step are needed for the readers to understand the significant role of water. Furthermore, it is confusing to me why water should be the source for protons during CO₂ reduction. Shouldn't it be HCO₃⁻, the strongest acid in this medium?

Reply and actions taken: First, we would like to explain that water other than HCO₃⁻ is the source of hydrogen (protons) for formate during the CO₂ reduction reaction over our catalyst. This is because the catalyst surface is negatively charged when a constant negative potential is applied during CO₂RR. Therefore, the positively charged K⁺(H₂O)_n cation rather than the negatively charged HCO₃⁻ anion is more likely to enter the double layer and participate in the discharge process. In particular, in our case, the surface S²⁻ species on the S-In catalyst surface could further promote the access of K⁺(H₂O)_n cation to the double layer of cathode surfaces via Coulomb interactions. The surface S²⁻ species would repel HCO₃⁻ anion.

To further clarify the hydrogen source of formate over the S2-In catalyst, we performed additional CO₂RR in 0.5 M KHCO₃/D₂O electrolyte by using D₂O to replace H₂O. After the reaction, the total amount of HCOO⁻ and DCOO⁻ in aqueous phase was quantified by HPLC. The amount of HCOO⁻ produced was determined by ¹H NMR. Then, the amount of DCOO⁻ was calculated by subtracting the amount of HCOO⁻ from the total amount of HCOO⁻ and DCOO⁻. The formation rates of HCOO⁻ and DCOO⁻ have been displayed in Supplementary Fig. 19. The result shows that, instead of HCOO⁻ (10 μmol h⁻¹ cm⁻²), DCOO⁻ (538 μmol h⁻¹ cm⁻²) is the overwhelming majority (Supplementary Fig. 19) in spite of 15000 μmol HCO₃⁻ present in the electrolyte (30 ml). Thus, we can conclude that the hydrogen in formate mainly originates from water but not HCO₃⁻. In addition, a higher formation rate of formate has been achieved in a hydrogen-free K₂SO₄ electrolyte than that in KHCO₃ electrolyte (Supplementary Fig. 21). This also implies that the hydrogen in formate does not come from HCO₃⁻.

Regarding the role of the activation of water in CO₂RR, we would first like to strengthen the following aspects already present in our previous manuscript. First, the activation of H₂O under alkaline conditions, which has been adopted in our present work, is a difficult step. Actually, it is known that the activation of H₂O in alkaline solution is also a difficult step for HER. Second, our experimental results by employing different cations (Na⁺, K⁺ and Cs⁺) in the electrolyte also indicate the importance of water activation for CO₂RR to formate over the S-In catalyst. By changing the cation from Na⁺ to K⁺ and further to Cs⁺ in the electrolyte, the formation rate of formate is significantly enhanced from 789 to 1002 and further to 1449 μmol h⁻¹ cm⁻² over the S2-In catalyst, whereas the activity only varies in a range of 490-580 μmol h⁻¹ cm⁻² over the S0-In catalyst without sulfur. For the HER reaction, it is widely accepted that the interaction between different hydrated cations and the double layer of electrode can be strengthened by catalyst modification (for example with sulfur), which significantly boosts the activation of H₂O (*Nat. Mater.* **15**, 197-203 (2016); *Chem* **3**, 122-133 (2017); *Science* **334**, 1256-1260 (2011)). We consider that the activity of CO₂RR over the S2-In catalyst is effectively promoted by the activation of H₂O, which is markedly influenced by the structure of hydrated cations. As compared to Na⁺(H₂O)₁₃ and K⁺(H₂O)₇, Cs⁺(H₂O)₆ with a smaller hydration number and a small radius of hydrated cation possesses a stronger interaction with S²⁻ sites on catalyst surfaces, hence providing higher activity for CO₂RR to formate.

To gain more direct information on the role of the activation of H₂O in CO₂RR, we have further performed studies on the kinetic isotope effect (KIE) of H/D for CO₂RR over the S2-In catalyst by using D₂O instead of H₂O. We quantified the

formation rates of HCOO⁻ and DCOO⁻ generated in electrolytes of H₂O and D₂O, respectively. The results have been displayed in Supplementary Fig. 19. We found that the substitution of hydrogen by deuterium in water led to a decline in the formation rate of formate. The KIE of H/D in CO₂RR was calculated to be 1.9. This KIE of H/D is characteristic of primary kinetic isotope effect (please see for example, *J. Phys. Chem. C* **114**, 3089-3097 (2010)). This result provide evidence that the dissociation of water is involved in the rate determining step for CO₂RR to formate over our S-In catalysts.

Regarding the reaction mechanism involving elementary steps for CO₂RR, we have proposed such a mechanism in Fig. 3e. The elementary steps for CO₂ to formate can be expressed more clearly as follow:

In fact, it is still controversial about whether the first electron is added to CO₂ to form CO₂^{*-} radical for CO₂RR (*Proc. Natl. Acad. Sci.* **114**, E8812-E8821 (2017)). Considering the very negative equilibrium potential for CO₂^{*-} formation (-1.9 V versus RHE) as compared to the potential actually employed for CO₂ reduction to formate (from -0.4 to -1.2 V versus RHE), we propose that the first electron is more likely to go to H₂O, and this bypasses the formation of high-energy CO₂^{*-} radical. The activated H^{*} generated from H₂O activation then reduces CO₂ to produce reaction intermediates, such as HCOO^{*}. This mechanism has also been supported by our DFT calculations in Fig. 3.

Based on the results and discussion described above, we have made the following revisions in the revised manuscript:

(1) To avoid misunderstanding, we have changed the following sentence in the previous manuscript: “As shown in Eq. 1, the activation of H₂O is also required for the conversion of CO₂ to formate, whereas so far this point has been overlooked in the CO₂RR” into “As shown in Eq. 1, the reduction of CO₂ to formate also consumes H₂O, but so far the activation of H₂O has been overlooked in the CO₂RR” (**please see Page 12, Lines 5-7**).

(2) We have expanded the discussion on the activation of H₂O under alkaline conditions. The following sentences have been added in the revised manuscript: “It is reported that the H₂ formation activity is one order of magnitude lower under alkaline conditions (pH = 13) than that in an acid electrolyte (pH = 1) during the HER over Au(111) surfaces³³, because of the difficulty in the reduction of water in alkaline electrolyte as compared to the discharging of hydronium in acid electrolyte. The

alkaline electrolyte is widely employed in literature for CO₂RR and also in our work. We consider that the activation of H₂O would also be a slow step for CO₂RR in alkaline medium” (please see Page 12, Lines 9-14).

(3) We have displayed the result of CO₂RR performed in D₂O solution in Supplementary Fig. 19. The description of the result and the discussion on the hydrogen resource of formate and the KIE of H/D have been added in the revised manuscript. Please see the following new paragraph: “To gain further insights into the role of the activation of H₂O in CO₂RR, we have conducted studies on the kinetic isotopic effect (KIE) of H/D over the S2-In catalyst. When D₂O was used to replace H₂O in 0.5 M KHCO₃ electrolyte, the formate formed was almost in the form of DCOO⁻ (538 μmol h⁻¹ cm⁻²) instead of HCOO⁻ (10 μmol h⁻¹ cm⁻²) (Supplementary Fig. 19). This indicates that the hydrogen in formate mainly originates from water rather than HCO₃⁻. The KIE of H/D in CO₂RR to formate was calculated to be 1.9. This KIE value is characteristic of primary kinetic isotopic effect³⁴. This result provides evidence that the dissociation of water is involved in the rate determining step for CO₂RR to formate over our S2-In catalyst” (please see from Page 12-Paragraph 3 to Page 13-Line 1).

(4) The experimental method for analyzing DCOO⁻ and HCOO⁻ formed during CO₂RR in D₂O solution has been added in Methods section as follows: “Electrocatalytic CO₂RR in D₂O solution was performed with similar procedures except for replacing H₂O with D₂O. For product analysis, the total amount of HCOO⁻ and DCOO⁻ in aqueous phase was quantified by HPLC after the reaction. The amount of HCOO⁻ produced was determined by ¹H NMR. Then, the amount of DCOO⁻ was calculated by subtracting the amount of HCOO⁻ from the total amount of HCOO⁻ and DCOO⁻” (please see Page 23, Paragraph 2).

(5) To make the mechanism described in Fig. 3e understood more easily, we have displayed the proposed elementary steps for CO₂RR to formate over the S2-In catalyst in Supplementary Scheme 1 (please see Supplementary Page 25) as follows:

The following sentence has also been added in the main text: “These proposed elementary steps are displayed in Supplementary Scheme 1” (please see Page 16, Paragraph 1, the last sentence).

Comment 3: It is unclear to me how the results in Fig S17 were obtained. Were they obtained under gas phase conditions? If this is the case, how can they reflect the

situation in an electrolyte medium?

Reply and actions taken: The CO₂ adsorption result displayed in Fig. S17 was obtained by gas-phase measurements. We agree with the reviewer that the measurement of CO₂ adsorption on an electrocatalyst, which is immersed in the liquid electrolyte medium, is a more convincing characterization method for CO₂RR study. However, this can hardly be achieved because of the difficulty in determining the CO₂ adsorption amount on electrocatalyst in the liquid medium. Thus, the adsorption of CO₂ is usually carried out under gas-phase conditions in the current studies for electrocatalytic CO₂RR (please see for examples: *Nature* **529**, 68-71 (2016); *Nat. Energy* **2**, 17087 (2017); *Nat. Commun.* **8**, 1828 (2017); *Sci. Adv.* **3**, 1701069 (2017)).

Nevertheless, it is generally believed that the adsorption of CO₂ measured under gas-phase conditions can provide information on the intrinsic properties of electrocatalysts, which are determined by the identity of metal surfaces, surface area, electronic structure and basicity of active surfaces. Although the absolute value of CO₂ adsorption amount under gas-phase conditions may be different from that in electrolyte medium, the variation trends for different catalysts, which are determined by the catalyst property, should be similar in the two cases. In other words, the variation trend in the capacity of electrocatalysts for CO₂ adsorption obtained under gas-phase conditions can reflect their behaviors for CO₂ adsorption in electrolyte medium.

Because of the reasons described above, we prefer to keep the CO₂ adsorption result (Supplementary Fig. 17) in the revised manuscript. Considering the comment raised by this reviewer, we have added the following sentence to explain that the CO₂ adsorption measurement was performed under gas-phase conditions: “*We performed CO₂ adsorption under gas-phase conditions to compare the CO₂ adsorption capacity among different catalysts*” (*please see Page 10, Lines 4-2 from bottom*).

Comment 4: It is unclear to me based on the DFT calculation why S doping promoted CO₂RR on In but not HER.

Reply and actions taken: Our DFT calculation result indicates that HER can be promoted by sulfur doping in the absence of CO₂ on In catalyst. The DFT calculation also suggests that the formation of HCOO*, the precursor of formate, can be preferentially formed in the presence of CO₂ and H₂O.

In more detail, our DFT calculation results displayed in Fig. 3b-Fig. 3d show that the Gibbs free energies for the reduction of CO₂ both to HCOO* and HCOOH* intermediates for formate formation (Fig. 3b) and to *COOH intermediate for CO formation (Fig. 3c) by the adsorbed H, which is formed from an electron/H₂O couple, have largely declined by doping sulfur onto In surface. The Gibbs free energy for the

activation of H₂O alone to H* species has also been found to decline significantly on S site of S–In surfaces (Fig. 3d). Because the activity of HER is known to relate to the Gibbs free energy of atomic hydrogen formed on the electrocatalyst surface (*J. Electrochem. Soc.* **153**, J23-J26 (2005)), the result in Fig. 3d indicates that the S modification accelerates HER on In catalyst and the S site is be the active site for H₂O activation. Our experimental result for HER in the absence of CO₂ (Supplementary Fig. 20) also confirms enhancement. In the presence of CO₂, our DFT calculation reveals that the doping with sulfur has turned the formation of HCOO*, the precursor of formate, on the S–In surface to be exergonic (Fig. 3b), whereas the formation of H* from H₂O alone and the formation of *COOH, the precursor of CO, still remain endergonic. We believe that this is the major reason for why the formation of formate but not the formation of H₂ is preferentially enhanced in the case of CO₂RR after the modification of In by sulfur (Fig. 1a), although sulfur can boost the activation of H₂O.

To make these points further clearer, we have rewritten one paragraph for the DFT calculation for HER on In and S-doped In surfaces as follows: “*We have further calculated the Gibbs free energies for the HER in the absence of CO₂ on pure In and sulfur-doped In surfaces. The formation energy of H* species is 0.21 eV on sulfur sites of sulfur-doped In, significantly lower than that on In sites of sulfur-doped In (0.69 eV) and pure In (0.82 eV) (Fig. 3d). The lower formation energy of H* species means a higher activity of H₂O dissociation on the electrocatalyst surface⁴⁰⁻⁴³. Therefore, our calculation results indicate that the sulfur modification can enhance the HER and the sulfur site on In surfaces is responsible for the dissociation of H₂O to form the adsorbed H* intermediate. The enhancement in HER in the absence of CO₂ by sulfur modification has also been confirmed by our experimental results (Supplementary Fig. 20). On the other hand, in the presence of CO₂, our DFT calculation reveals that the doping with sulfur has turned the formation of HCOO*, the precursor of formate, on the S–In surface to be exergonic (Fig. 3b), whereas the formation of H* from H₂O alone and the formation of *COOH, the precursor of CO, still remain endergonic. We believe that this is the major reason for why the formation of formate but not the formation of H₂ is preferentially enhanced in the case of CO₂RR after the modification of In by sulfur (Fig. 1a), although sulfur can boost the activation of H₂O” (please see Page 15, Paragraph 2).*

Comment 5: In supplementary Fig. 10, why is the S0-In has a larger interplanar spacing than the other materials?

Reply and actions taken: We thank the reviewer for pointing out our statistical errors. We have checked and re-measured more HRTEM images for particles in the S0-In

sample. Now, we confirm that the average interplanar spacings of (101) facet of the S0-In sample is 0.272 nm. We have revised the value in Supplementary Fig. 10a (*please see Supplementary Information, Page 9*).

Response to Reviewer 2

General Comments: The authors made a considerable effort to modulate the main messages and to provide additional experimental evidence supporting their main claims.

To my understanding, the main claim of sulfur acting as a promoter of the activity by a mechanism where water may act as the proton source is now solid enough to be considered seriously by the research community, and thus valid to inspire new work in multicomponent electrocatalysts for this and other reactions. However, at this point, authors should discuss the very recently published experimental-theoretical study on the mechanism behind the promotion of the formate route by sulfur species on Cu by Pérez-Ramírez et al. (J. Phys. Chem. Lett. 2018, doi: 10.1021/acs.jpcllett.8b03212), which suggests a very different mechanism to the one proposed by the authors.

Additional experiments performed by the authors are consistent with the overall picture, which points out to the strong tendency of sulfur to promote the formate route almost independently of the structure and chemical state of the base In. Authors also made the effort to shed light on the importance of oxidic In species on the reaction, though failed to consider them in the DFT study. In general, even though the picture is still fragmented, their findings are robust and have the potential to act as strong seeds for further developments in terms of novel catalysts and understanding.

I, therefore, recommend publication upon consideration of previous comments.

Reply and actions taken: We appreciate the positive evaluation by this reviewer to our revised manuscript. We have read with interest the paper provided by the reviewer, which has been published very recently, and have cited this paper as reference 44 in our revised manuscript. We agree with the reviewer that the addition of the discussion on the promoting mechanism of sulfur on Cu surfaces, which is different from that proposed in the present work, can improve the completeness of our manuscript.

Therefore, we have added the discussion on the promoting effect of sulfur in Cu-catalyzed CO₂RR reported by Pérez-Ramírez and co-workers. We have also discussed the major difference in the roles of sulfur between Cu-based catalytic system and our In-based catalytic system. The following new paragraph has been added in the revised manuscript: “*It is noteworthy that Pérez-Ramírez and co-workers*

recently reported a promotion effect of sulfur modification on the reduction of CO₂ to formate over Cu catalyst^{21,44}. Unlike In catalyst alone (S0–In) with high FE of formate (85%) (Fig. 1a), the Cu catalyst alone showed higher FEs of CO and H₂. The doping of sulfur mainly changed the product selectivity and the FE of formate increased from 26% to 78% after sulfur modification over the Cu catalyst. The sulfur adatom on Cu surfaces is proposed to participate actively in CO₂RR as a nucleophile either by transferring a hydride or by tethering CO₂, thus suppressing the formation of CO.⁴⁴ The different behaviors of sulfur doping on In and Cu catalysts reveal diversified functioning mechanisms of sulfur for CO₂RR” (please see Page 16, Paragraph 2).

REVIEWERS' COMMENTS:

Reviewer #1 (Remarks to the Author):

I don't think the arguments made the authors are convincing. For example, the electrolysis results obtained from the 0.5 M KHCO₃/D₂O medium were used to confirm the source of proton. This is incorrect since H⁺ and D⁺ can exchange in the medium to form DCO₃⁻. However, I think the authors have done a good job by providing substantial amount of new information and carefully described experimental details. Therefore, the revised version is acceptable for publication in my view.

Responses to Reviewers

Response to Reviewer 1

Comments: I don't think the arguments made the authors are convincing. For example, the electrolysis results obtained from the 0.5 M KHCO₃/D₂O medium were used to confirm the source of proton. This is incorrect since H⁺ and D⁺ can exchange in the medium to form DCO₃⁻. However, I think the authors have done a good job by providing substantial amount of new information and carefully described experimental details. Therefore, the revised version is acceptable for publication in my view.

Reply and actions taken: We appreciate the comments raised by Reviewer 1. However, we believe that the source of proton for formate formation is water but not HCO₃⁻. First, HCO₃⁻ is an anion, while the catalyst surface is negatively charged when a constant negative potential is applied during CO₂RR. Therefore, the negatively charged HCO₃⁻ anion is unlikely to enter the double layer and participate in the discharge process due to the electrostatic repulsion, and thus cannot be a proton donor for formate formation. Second, when hydrogen-free K₂SO₄ was used as the electrolyte instead of KHCO₃, the rate of formate formation became rather higher (Supplementary Fig. 19). This also implies that the hydrogen in formate does not come from the electrolyte. Moreover, we have also measured the kinetic isotope effect (KIE) of H/D in K₂SO₄ electrolyte and obtained the same H/D KIE value (1.9) with that in KHCO₃ electrolyte (The result has been added in Supplementary Fig. 19), indicating that the activation of water is an important step for CO₂RR.

In the revised manuscript, we have added the following two sentences to describe the result for KIE of H/D in K₂SO₄ electrolyte: “*We have also measured KIE of H/D in 0.5 M K₂SO₄ electrolyte and obtained the same result (Supplementary Fig. 19).*” (*please see Page 13, Paragraph 1*).